# TimePre: Bridging Accuracy, Efficiency, and Stability in Probabilistic Time-Series Forecasting

## Abstract

We propose **TimePre**, a simple framework that unifies the efficiency of Multilayer Perceptron (MLP)-based models with the distributional flexibility of Multiple Choice Learning (MCL) for Probabilistic Time-Series Forecasting (PTSF). **Stabilized Instance Normalization (SIN)**, the core of TimePre, is a normalization layer that explicitly addresses the trade-off among accuracy, efficiency, and stability. SIN stabilizes the hybrid architecture by correcting channel-wise statistical shifts, thereby preventing the hypothesis collapse that otherwise destabilizes this combination. Extensive experiments on six benchmark datasets demonstrate that TimePre achieves the best Distortion on all six benchmarks and the best or highly competitive CRPS-Sum on most benchmarks. Critically, TimePre achieves inference speeds that are orders of magnitude faster than sampling-based models, and is more stable than prior MCL approaches.

## 1 Introduction

Probabilistic time-series forecasting (PTSF) aims to model the conditional distribution of future trajectories given historical observations (Kim et al., 2025). It is important in a wide range of applications, including weather prediction (Wu et al., 2023), energy management (Rajagukguk et al., 2020), and finance. Recent advances have introduced several generative paradigms for PTSF (LeCun et al., 2015; Hu et al., 2025; Wang et al., 2019), including diffusion-based models such as TimeGrad (Rasul et al., 2021), flow-based models such as TempFlow (Rasul et al., 2020), and copula-based approaches such as TACTiS-2 (Ashok et al., 2024).

However, these approaches typically rely on costly multi-step sampling to represent uncertainty, which limits their efficiency and practical scalability. To address this issue, TimeMCL (Cortes et al., 2025; Perera et al., 2024a) introduced a non-sampling formulation for PTSF based on Multiple Choice Learning (MCL), modeling uncertainty with a finite set of discrete hypotheses within an autoregressive RNN (Lyu et al., 2021; D YAMADA et al., 2021). In this framework, each prediction head is trained under a winner-takes-all (WTA) objective, where only the hypothesis with the smallest loss for each sample receives gradient updates. While this competitive mechanism encourages specialization among hypotheses, it also induces highly uneven gradient allocation (Rodriguez Domínguez et al., 2025), often resulting in limited hypothesis diversity and unstable optimization.

At the same time, recent progress in long-term time-series forecasting (LTSF) has shown that lightweight

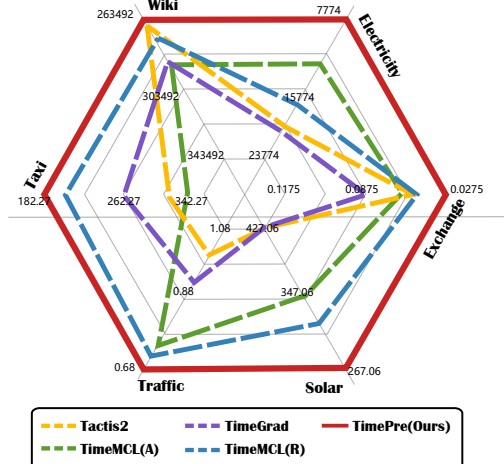

Figure 1: Model performance comparison on the Distortion metric across six real-world benchmark datasets. Lower is better.

architectures, such as linear models and multilayer perceptron (MLP)-based backbones (Rumelhart et al., 1986), can outperform more complex Transformer-based methods (Nie et al., 2023; Zhou et al., 2021) in both accuracy and efficiency. Representative examples include DLinear (Zeng et al., 2023), TiDE (Das et al.,

2023), and TimeMixer (Wang et al., 2024). These results suggest that simple forecasting backbones can be highly effective when properly designed.

Motivated by this trend, a natural question is whether the efficiency of lightweight forecasting backbones can be combined with the distributional flexibility of the MCL paradigm. We find that naively combining the two leads to severe optimization instability: training becomes unstable and hypotheses quickly collapse. Importantly, our controlled analysis (Section 4.4) indicates that this failure is *not* intrinsic to lightweight backbones or to the winner-takes-all (WTA) objective in isolation; rather, it emerges from their interaction with inadequate input normalization, and it can be largely recovered once a suitable reversible normalization is introduced. The underlying mechanism is a scale-sensitivity problem: linear projections lack the implicit regularization and manifold constraints provided by nonlinear encoders such as LSTMs (Hochreiter & Schmidhuber, 1997; Neyshabur et al., 2017; Zhang et al., 2021; 2025; 2018), so scale disparities in real-world data are directly exposed and can be rapidly amplified during optimization. Under the competitive WTA objective, only a small subset of scale-aligned hypotheses then tends to receive consistent gradient updates, while the others stagnate and collapse.

To address this issue, we propose **TimePre**, a probabilistic forecasting framework built on Stabilized Instance Normalization (SIN) and a direct multi-hypothesis predictor. SIN performs adaptive channel-wise rescaling before the input reaches the lightweight encoder, mitigating statistical shifts (Quionero-Candela et al., 2009) and improving optimization stability under the WTA objective. This design preserves hypothesis diversity while maintaining the efficiency advantages of direct forecasting. As shown in Figure 1, TimePre achieves strong forecasting performance across six benchmark datasets while being substantially faster than existing sampling-based methods.

Our main contributions are summarized as follows:

- We identify and diagnose a severe optimization instability that arises when the MCL paradigm is combined with lightweight forecasting backbones, and show through a controlled isolation study that it stems from the interaction between channel-wise scale imbalance and inadequate normalization, rather than from any single component in isolation.

- We propose TimePre, which combines the efficiency of lightweight forecasting models with the distributional modeling capability of MCL. Its core component, Stabilized Instance Normalization, improves optimization stability and supports balanced competition among hypotheses. We additionally give a theoretical account, under a tractable quantization surrogate, of how SIN's robust normalization helps keep more hypotheses active (Appendix E).

- Extensive experiments on six benchmark datasets show that TimePre achieves state-of-the-art or highly competitive performance across probabilistic forecasting metrics, while offering substantially faster inference than sampling-based baselines.

## 2 Related Work

### 2.1 Multiple Choice Learning and Multi-Hypothesis Forecasting

Multiple Choice Learning (MCL) models diverse outcomes under uncertainty. Introduced as an assignment-based multi-model training scheme by Guzman-Rivera et al. (2012) and later reformulated as a differentiable winner-takes-all (WTA) objective for multi-head networks (Rupprecht et al., 2017), it lets different heads specialize in different modes of the target distribution. From a vector-quantization perspective, MCL learns a finite set of representative codevectors that approximate a conditional distribution (Gersho & Gray, 1992; Loubes & Pelletier, 2017; Letzelter et al., 2024).

A central challenge is the instability of hard competition: since only the winning head receives dominant gradients, training is sensitive to initialization and prone to mode collapse. Relaxed or stabilized WTA variants address this through annealed or stochastic formulations (Lee et al., 2016; Perera et al., 2024a) and learned scoring or resilient assignment mechanisms (Perera et al., 2024b). In probabilistic forecasting, TimeMCL adapts MCL to generate a discrete set of plausible trajectories (Cortes et al., 2025), offering an attractive trade-off among forecast quality, diversity, and efficiency. However, existing MCL forecasters rely on autoregressive or heavier sequential backbones, and their interaction with lightweight linear backbones remains underexplored.

## 2.2 Time-Series Forecasting Backbones

Time-series forecasting has evolved from statistical and machine-learning methods (Ho & Xie, 1998; Friedman, 2000) to deep architectures. Early probabilistic models such as DeepAR combine autoregressive recurrent backbones (GRU/LSTM) with parametric likelihoods for uncertainty estimation (Salinas et al., 2020; Kim et al., 2025), but recurrence limits parallelism and long-horizon efficiency. Transformer-based models such as Informer, Autoformer, and FEDformer instead use self-attention or frequency-domain modules to capture long-range dependencies (Vaswani et al., 2017; Zhou et al., 2021; Wu et al., 2021; Zhou et al., 2022), although later studies attribute much of their gain to design choices such as normalization, decomposition, and scaling rather than attention alone. Closely related to our approach, RevIN standardizes each series with instance-specific statistics and reverses the transformation at the output (Kim et al., 2022). SIN follows this reversible design, but replaces the standard mean and variance with trimmed estimates to improve robustness to outliers and scale spikes under WTA training.

This motivated lightweight linear and MLP-style models such as DLinear, TiDE, and TimeMixer, which reach strong performance at far lower cost (Zeng et al., 2023; Das et al., 2023; Wang et al., 2024). Most such models, however, target deterministic prediction rather than multi-hypothesis probabilistic forecasting. Our method addresses how to combine their efficiency with the distributional flexibility of MCL while mitigating training instability.

## 3 Approach

### 3.1 Preliminaries

We consider a multivariate stochastic process $\{x_t\}_{t=1}^T$, where $x_t \in \mathbb{R}^D$ denotes a $D$-dimensional observation at time $t$. Given a look-back window of length $L$ and a forecast horizon of length $H$, we define the input–output pair as $\mathbf{X}_t = [x_{t-L}, \ldots, x_{t-1}]^\top \in \mathbb{R}^{L \times D}$ and $\mathbf{Y}_t = [x_t, \ldots, x_{t+H-1}]^\top \in \mathbb{R}^{H \times D}$.

The forecasting objective is to learn a mapping $f_\Theta : \mathbb{R}^{L \times D} \to \mathbb{R}^{H \times D}$ that minimizes the conditional risk

$$\min_\Theta \ \mathbb{E}_{(\mathbf{X},\mathbf{Y}) \sim \mathcal{D}} \big[ \ell(f_\Theta(\mathbf{X}), \mathbf{Y}) \big], \tag{1}$$

where $\ell(\cdot, \cdot)$ is a task-specific loss induced by the forecasting likelihood and $\mathcal{D}$ denotes the underlying data distribution.

However, the objective in Equation (1) defines a deterministic mapping, whereas real-world temporal processes are often stochastic and may admit multiple plausible futures. A single predictor is therefore insufficient to approximate the full conditional distribution $p(\mathbf{Y} \mid \mathbf{X})$. To model such multi-modality in a tractable way, we adopt the functional quantization formulation (Guzman-Rivera et al., 2012), in which a finite set of $K$ hypothesis functions $\{f_\Theta^{(k)}\}_{k=1}^K$ jointly approximates the conditional manifold by minimizing

$$\min_{\{f_\Theta^{(k)}\}} \mathbb{E}_{(\mathbf{X},\mathbf{Y})} \left[ \min_{k=1,\ldots,K} d\big(f_\Theta^{(k)}(\mathbf{X}), \mathbf{Y}\big) \right], \tag{2}$$

where $d(\cdot, \cdot)$ denotes a trajectory-level discrepancy measure, typically the $\ell_2$ distance. Under this view, each $f_\Theta^{(k)}$ serves as a representative centroid of a distinct mode of $p(\mathbf{Y} \mid \mathbf{X})$, providing a bridge between deterministic regression and probabilistic forecasting.

This formulation follows the multi-hypothesis learning paradigm introduced in Multiple Choice Learning (Guzman-Rivera et al., 2012; Lee et al., 2016) and later extended to structured prediction (Rupprecht et al., 2017). Probabilistic forecasting approaches such as DeepAR (Salinas et al., 2020), MQRNN (Wen et al., 2017), and Deep Ensembles (Lakshminarayanan et al., 2017) are also related in that they approximate $p(\mathbf{Y} \mid \mathbf{X})$ through multiple predictive hypotheses, although they differ substantially in training and inference mechanisms.

### 3.2 Diagnosis: Instability of Linear Backbones under MCL

Our empirical study reveals a consistent instability when modern linear backbones are combined with the multi-hypothesis formulation in Equation (2). Rather than causing numerical divergence, this combination

tends to produce learning stagnation and hypothesis collapse (Rupprecht et al., 2017), where only a small subset of heads remains active while the others fail to learn meaningful predictions. We attribute this behavior to the interaction of two factors: inter-variable scale imbalance in real-world multivariate data, and the absence of implicit regularization in linear mappings.

**Impact of scale imbalance on the WTA objective.** Real-world multivariate time-series, such as Electricity and Wiki, contain variables with substantially different physical units and magnitudes. Standard global normalization, such as dataset-level $z$-score normalization, is often insufficient in this setting because it does not correct per-instance, per-variable scale imbalance. This is particularly problematic under the winner-takes-all (WTA) objective in Equation (2), which is highly sensitive to relative error scale. A hypothesis $f_\Theta^{(k)}$ that is initialized slightly closer to a high-magnitude variable may repeatedly win the arg min operation, even if it performs poorly on other variables. As a result, gradient updates become concentrated on a small subset of heads, while the remaining hypotheses receive little useful supervision. This imbalance leads to unstable optimization and eventually to hypothesis collapse.

**Linear backbones as unconstrained amplifiers.** This failure mode becomes more severe when the forecasting backbone is linear. Nonlinear encoders such as LSTMs (Hochreiter & Schmidhuber, 1997) and Transformers (Vaswani et al., 2017) provide a form of implicit regularization by coupling features through shared nonlinear representations. In contrast, linear backbones directly propagate input-scale disparities and initialization bias into the WTA competition. Without a shared manifold to regularize head behavior, the $K$ hypotheses compete in a poorly conditioned feature space, and small initial differences can quickly grow into large functional disparities. This effect makes inactive heads difficult to recover and impedes stable optimization.

**Requirements for stabilization.** This diagnosis suggests that a stabilization mechanism for linear MCL forecasting should satisfy three requirements:

1. **Per-instance, per-variable operation.** It should correct inter-variable scale imbalance at the channel level, rather than relying on batch-level statistics.
2. **Robustness to non-stationarity and outliers.** It should remain stable under spikes, distribution shift, and heavy-tailed temporal observations.
3. **Analytical reversibility.** It should allow predictions to be mapped back to the original data scale without losing physical interpretability.

To satisfy these requirements, we introduce Stabilized Instance Normalization (SIN), a robust, channel-wise, and reversible preconditioning layer for multi-hypothesis forecasting with lightweight backbones.

### 3.3 TimePre

**Overall architecture.** TimePre is a three-stage pipeline designed to combine the computational efficiency of lightweight forecasting backbones with the probabilistic output structure of Multiple Choice Learning. Given an input context window $\mathbf{X} \in \mathbb{R}^{L \times D}$, TimePre generates $K$ candidate future trajectories $\{\widehat{\mathbf{Y}}^{(k)}\}_{k=1}^{K}$ through three components: Stabilized Instance Normalization $\phi$, a linear temporal encoder Enc, and a multi-hypothesis decoder Dec. The overall pipeline is $\{\widehat{\mathbf{Y}}^{(1)}, \ldots, \widehat{\mathbf{Y}}^{(K)}\} = \text{Dec}\big(\text{Enc}(\phi(\mathbf{X}))\big)$.

Here, $\phi$ preconditions the non-stationary input by reducing the scale imbalance diagnosed in Section 3.2. The stabilized representation is then processed by the linear encoder and finally mapped to a set of diverse trajectory hypotheses by the decoder. This design follows the multi-hypothesis forecasting formulation of (Guzman-Rivera et al., 2012; Lee et al., 2016; Rupprecht et al., 2017; Perera et al., 2024b), while replacing heavy recurrent or Transformer-based backbones with a lightweight encoder and a reversible normalization step.

**Stabilized Instance Normalization (SIN).** SIN is introduced as a minimal preconditioning mechanism to stabilize WTA optimization under lightweight linear backbones. Its goal is not to increase model expressiveness, but to provide robust, channel-wise, and reversible normalization. The reversible per-instance normalization that SIN adopts builds on RevIN (Kim et al., 2022), which introduced reversible instance normalization for forecasting under distribution shift; SIN extends it with a robust, trimmed estimator of the per-channel statistics, which we motivate next. Standard Instance Normalization (Ulyanov et al., 2016) satisfies the

per-instance requirement, but it is sensitive to outliers and distribution shift, both of which are common in non-stationary time-series. A single extreme observation can substantially distort the empirical mean and variance, which in turn destabilizes normalization. To improve robustness, SIN computes channel-wise statistics using a trimmed estimator. For each variable (channel) $d$, let $\mathbf{x}^{(d)} = (s_1^{(d)}, \ldots, s_L^{(d)})^\top \in \mathbb{R}^L$, and let $s_{(i)}^{(d)}$ denote its $i$-th order statistic. Given a trimming ratio $p \in [0, 0.5)$, we set $k = \lfloor pL \rfloor$ and compute the robust mean and variance over the central $L - 2k$ values:

$$\mu_r^{(d)} = \frac{1}{L - 2k} \sum_{i=k+1}^{L-k} s_{(i)}^{(d)}, \quad v_r^{(d)} = \frac{1}{L - 2k} \sum_{i=k+1}^{L-k} \left( s_{(i)}^{(d)} - \mu_r^{(d)} \right)^2, \quad \sigma_r^{(d)} = \sqrt{v_r^{(d)} + \epsilon}. \tag{3}$$

The normalization and its exact inverse are then applied to the original, unsorted sequence:

$$\tilde{\mathbf{x}}^{(d)} = \frac{\mathbf{x}^{(d)} - \mu_r^{(d)} \mathbf{1}_L}{\sigma_r^{(d)}}, \qquad \mathbf{x}^{(d)} = \tilde{\mathbf{x}}^{(d)} \sigma_r^{(d)} + \mu_r^{(d)} \mathbf{1}_L. \tag{4}$$

Because SIN operates independently on each channel and each instance, it directly addresses the scale imbalance that distorts the WTA competition. At the same time, the use of trimmed statistics improves robustness under non-stationarity, while the closed-form inverse preserves interpretability in the original data space. Sections 4.3 and 4.4 show empirically that this robust design, rather than reversible normalization alone, is what stabilizes multi-hypothesis training, and Appendix E gives a theoretical account of why trimming helps.

**Linear temporal encoder.** Following recent linear-centric forecasting models (Zeng et al., 2023), we use a simple linear layer as the temporal encoder. For each normalized channel $\tilde{\mathbf{x}}^{(d)} \in \mathbb{R}^L$, the encoder projects the look-back window directly to the forecast horizon, $\mathbf{z}^{(d)} = W^{(d)} \tilde{\mathbf{x}}^{(d)} + b^{(d)}$, with $W^{(d)} \in \mathbb{R}^{H \times L}$ and $b^{(d)} \in \mathbb{R}^H$. The latent representation is then formed as $\mathbf{Z} = [\mathbf{z}^{(1)} \mid \cdots \mid \mathbf{z}^{(D)}] \in \mathbb{R}^{H \times D}$. This encoder is computationally efficient and preserves the channel-wise forecasting structure used in lightweight time-series models.

**Multi-hypothesis decoder.** The decoder contains $K$ parallel prediction heads. Each head consists of two components:

1. a trajectory head $f_\theta^{(k)}$ that predicts a trajectory hypothesis $\widehat{\mathbf{Y}}^{(k)} \in \mathbb{R}^{H \times D}$ from the shared latent representation $\mathbf{Z}$; and

2. a confidence head $g_\theta^{(k)}$ that estimates the confidence score $\gamma^{(k)}$ of the corresponding hypothesis.

Formally, $\widehat{\mathbf{Y}}^{(k)} = f_\theta^{(k)}(\mathbf{Z})$ and $\gamma^{(k)} = \sigma(g_\theta^{(k)}(\mathrm{vec}(\mathbf{Z})))$, where $\mathrm{vec}(\cdot)$ denotes vectorization and $\sigma(\cdot)$ is the sigmoid function.

### 3.4 Training and Inference

For a training pair $(\mathbf{X}, \mathbf{Y})$, we first compute the per-head reconstruction loss $\mathcal{L}^{(k)} = \frac{1}{HD} \big\| \widehat{\mathbf{Y}}^{(k)} - \mathbf{Y} \big\|_F^2$, and define the winning hypothesis as $k^* = \arg\min_k \mathcal{L}^{(k)}$.

**Relaxed WTA objective.** To mitigate hypothesis starvation, we adopt an $\varepsilon$-relaxed WTA objective (Rupprecht et al., 2017; Perera et al., 2024b; Seo et al., 2020). This objective assigns most of the gradient to the winner while preserving a smaller training signal for the remaining heads:

$$\mathcal{L}_{\text{R-WTA}} = (1 - \varepsilon) \mathcal{L}^{(k^*)} + \frac{\varepsilon}{K - 1} \sum_{j \neq k^*} \mathcal{L}^{(j)}. \tag{5}$$

**Confidence calibration.** The confidence scores $\gamma^{(k)}$ are trained to identify the winning hypothesis. We use a binary cross-entropy objective in which the winner is treated as the positive class:

$$\mathcal{L}_{\text{score}} = -\frac{1}{K} \left[ \log \gamma^{(k^*)} + \sum_{k \neq k^*} \log \big( 1 - \gamma^{(k)} \big) \right]. \tag{6}$$

**Total objective.** The final objective combines the relaxed WTA loss and the calibration loss:

$$\min_{\Theta} \; \mathbb{E}_{(\mathbf{X},\mathbf{Y})} \left[ \mathcal{L}_{\text{R-WTA}} + \beta \, \mathcal{L}_{\text{score}} \right], \tag{7}$$

where $\beta$ controls the trade-off between trajectory accuracy and confidence calibration.

**Inference.** At inference time, the model performs a single forward pass to produce all $K$ hypotheses $\{\widehat{\mathbf{Y}}^{(k)}\}$ and their confidence scores $\{\gamma^{(k)}\}$. Together, these outputs form the final probabilistic forecast. Under squared-error distortion, the framework can also be interpreted as learning a soft Voronoi partition of future trajectory space, where each hypothesis head approximates the centroid of a conditional mode of $p(\mathbf{Y} \mid \mathbf{X})$.

## 4 Experiment

### 4.1 Experimental Setup

**Datasets.** We follow the standard evaluation protocol in multivariate probabilistic forecasting and benchmark our model on six well-established real-world benchmark data sets from the GluonTS library (Alexandrov et al., 2019). These data sets cover multiple domains, including energy (Wu et al., 2019), finance (Lai et al., 2017), and transportation (Li et al., 2018). All data are preprocessed following prior work to ensure fair and consistent comparison across baselines. Full dataset statistics and per-dataset descriptions are provided in Appendix B.

**Metrics.** We follow the standard evaluation protocol used in prior MCL-based probabilistic forecasting work (Cortes et al., 2025) and report four metrics (formally defined in Appendix C): Distortion (Lee et al., 2016), CRPS-Sum (Gneiting & Raftery, 2007), FLOPs, and runtime. Distortion is the primary metric, measuring the mean Euclidean distance between each target sequence and its closest predicted hypothesis under the winner-takes-all objective:

$$D_2 = \frac{1}{N} \sum_{i=1}^{N} \min_{k=1,\dots,K} d\big( \mathcal{F}_{\theta}^{k}(x_{1:t_0-1}^{i}), \, x_{t_0:T}^{i} \big), \tag{8}$$

where $K$ is the number of hypotheses and $N$ is the number of test samples. This metric directly evaluates hypothesis coverage and is therefore well suited for MCL-based forecasting. CRPS-Sum complements Distortion by assessing the overall quality of probabilistic forecasts via the distance between the predicted distribution and the ground truth, computed for hypothesis-based forecasts as described in Appendix A.3.

**Baselines.** We compare with six representative probabilistic forecasting methods covering a wide range of paradigms, resulting in eight baseline variants (described in Appendix D). The selected baselines include ETS (Hyndman et al., 2002), DeepAR (Salinas et al., 2020), TimeGrad (Rasul et al., 2021), TempFlow (Rasul et al., 2020), Tactis2 (Ashok et al., 2024), and TimeMCL (Cortes et al., 2025). Among them, TempFlow is implemented in two versions based on LSTM and Transformer backbones. To evaluate multi-hypothesis forecasting methods under consistent conditions, we additionally include two WTA-based training variants, Relaxed-WTA and Annealed-MCL.

**Training Details.** All models follow the training protocol of prior MCL-based probabilistic forecasting work: Adam optimization, random-window sampling, fixed batch size, and early stopping. Unless otherwise stated, all main benchmark results are reproduced in our environment and reported over five random seeds. All reproduced results, diagnostic experiments, and efficiency measurements are run under the same software environment and hardware platform. Full TimePre hyperparameters, including optimizer settings, training length, batch size, and early-stopping patience, are provided in Appendix A. Runtime and FLOPs measurement details are provided in Appendix A.2.

**Hyperparameter tuning protocol.** To avoid per-dataset tuning bias, all methods use fixed configurations. For baseline methods, including ETS, DeepAR, TempFlow, TempFlow (Trf.), TimeGrad, Tactis2, TimeMCL (R.), and TimeMCL (A.), we follow the public TimeMCL reproduction scripts and the recommended settings of the corresponding original implementations. TimePre uses one default configuration across all six benchmarks, with no per-dataset tuning; the exact values are listed in Table 11. The sensitivity analyses in Figure 6 show that this default is at or near the best grid setting, suggesting that the reported gains do not stem from dataset-specific hyperparameter search.

Table 1: Distortion risk under 16 hypotheses. We report mean ± standard deviation over five random seeds. The best results are in **bold**, and the second-best results are underlined. Lower is better.

| Model | Electricity | Exchange | Solar | Traffic | Taxi | Wiki |
|---|---|---|---|---|---|---|
| ETS | 23590 ± 2474 | 0.0796 ± 0.0030 | 692.32 ± 22.16 | 2.73 ± 0.02 | 609.67 ± 1.89 | 835095 ± 37871 |
| TempFlow (Trf.) | 17521 ± 2691 | 0.1150 ± 0.0290 | 466.25 ± 23.57 | 1.38 ± 0.06 | 308.62 ± 21.75 | 561226 ± 26593 |
| Tactis2 | 13972 ± 917 | 0.0396 ± 0.0026 | 405.74 ± 17.19 | 0.87 ± 0.02 | 243.63 ± 9.10 | 263975 ± 11178 |
| TimeGrad | 14255 ± 1682 | 0.0576 ± 0.0090 | 406.91 ± 16.08 | 0.83 ± 0.02 | 221.32 ± 7.37 | 275437 ± 2645 |
| DeepAR | 184424 ± 19957 | 0.1320 ± 0.0204 | 865.61 ± 36.02 | 2.55 ± 0.12 | 477.93 ± 15.22 | 382340 ± 6592 |
| TempFlow | 17429 ± 1131 | 0.1168 ± 0.0325 | 424.24 ± 15.91 | 1.33 ± 0.02 | 293.76 ± 17.29 | 395996 ± 21535 |
| TimeMCL (R.) | 12693 ± 1772 | 0.0380 ± 0.0025 | 292.15 ± 11.68 | 0.71 ± 0.01 | 191.23 ± 5.34 | 268832 ± 9439 |
| TimeMCL (A.) | 10335 ± 767 | 0.0443 ± 0.0051 | 308.16 ± 14.87 | 0.72 ± 0.02 | 252.84 ± 30.62 | 276315 ± 9782 |
| TimePre (Ours) | **7774 ± 203** | **0.0275 ± 0.0004** | **267.06 ± 1.55** | **0.68 ± 0.02** | **182.27 ± 1.86** | **263492 ± 2368** |

Table 2: CRPS-Sum comparison on six benchmark datasets. Values are multiplied by 100. We report mean ± standard deviation. The best results are in **bold**, and the second-best results are underlined. Lower is better.

| Model | Electricity | Exchange | Solar | Traffic | Taxi | Wiki |
|---|---|---|---|---|---|---|
| ETS | 8.80 ± 0.95 | 1.27 ± 0.14 | 58.00 ± 1.67 | 22.57 ± 0.43 | 90.85 ± 0.35 | 15.80 ± 0.77 |
| TempFlow (Trf.) | 7.82 ± 1.79 | 2.30 ± 0.66 | 56.98 ± 4.68 | 49.84 ± 1.35 | 44.74 ± 4.33 | 17.85 ± 2.50 |
| Tactis2 | 5.36 ± 0.34 | 0.82 ± 0.15 | 40.58 ± 2.48 | 13.19 ± 1.24 | 22.52 ± 1.60 | 6.24 ± 0.87 |
| TimeGrad | 4.56 ± 0.33 | 1.39 ± 0.38 | **37.30 ± 1.39** | **5.89 ± 0.21** | **16.88 ± 2.17** | 8.08 ± 0.95 |
| DeepAR | 117.42 ± 11.62 | 2.86 ± 0.56 | 154.59 ± 25.39 | 96.25 ± 15.93 | 100.16 ± 1.95 | 80.25 ± 11.73 |
| TempFlow | 7.08 ± 0.62 | 2.65 ± 0.91 | 52.64 ± 3.28 | 49.15 ± 1.15 | 44.55 ± 6.93 | 14.49 ± 2.00 |
| TimeMCL (R.) | 5.46 ± 0.85 | 1.05 ± 0.12 | 41.12 ± 4.23 | 8.68 ± 1.10 | 46.19 ± 11.79 | 14.50 ± 3.84 |
| TimeMCL (A.) | 4.50 ± 0.43 | 1.32 ± 0.29 | 40.62 ± 5.88 | 8.21 ± 0.95 | 61.43 ± 12.85 | 17.23 ± 4.10 |
| TimePre (Ours) | **3.15 ± 0.32** | **0.72 ± 0.03** | 39.79 ± 0.66 | 11.81 ± 1.22 | 21.72 ± 0.53 | **6.14 ± 0.24** |

## 4.2 Main Results

Tables 1 and 2 summarize the quantitative results of Distortion and CRPS-Sum, comparing the proposed TimePre with baseline models under 16 hypotheses. Figure 2 illustrates the computation–performance trade-off on the Exchange data set. Figure 3 provides qualitative comparisons among TimePre, TimeMCL (R.), and TimeMCL (A.).

**Distortion.** Table 1 shows that TimePre achieves the best Distortion on all six data sets. It achieves a 38.8% reduction compared to TimeMCL (R.) on Electricity and a 27.6% improvement on Exchange. Similar trends are observed on Solar and Taxi. On Wiki, TimePre still achieves the lowest Distortion, though the margin over the second-best Tactis2 is relatively small (263492 vs. 263975). In addition to accuracy, TimePre exhibits notably lower variance across runs (e.g., ±203 on Electricity vs. ±1772 for TimeMCL (R.)).

**CRPS-Sum.** Table 2 reports CRPS-Sum for all nine methods. TimePre is best on Electricity, Exchange, and Wiki and second on Solar and Taxi, and is the strongest MCL-based method except on Traffic. TimeGrad attains the lowest CRPS-Sum on the remaining datasets, consistent with CRPS-Sum rewarding full-distribution calibration, where iterative sampling models retain an edge at higher inference cost.

**Computational Cost.** To evaluate computational cost, we measure both runtime and FLOPs on the Exchange data set. As shown in Figure 2, TimePre avoids iterative autoregressive sampling and generates all future predictions in a single forward pass, resulting in strong inference efficiency. Its inference time is the fastest among all compared models, taking only 0.03s per batch. In terms of computational load, TimePre requires $4.28 \times 10^4$

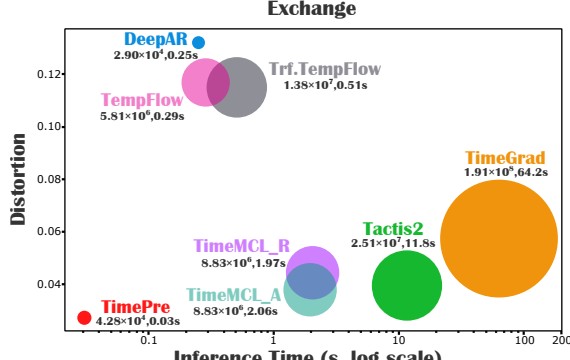

Figure 2: Computation–performance trade-off on the Exchange data set under 16 hypotheses. Lower is better. Circle size indicates FLOPs.

FLOPs, second only to DeepAR ($2.90 \times 10^4$ FLOPs), while achieving substantially lower distortion.

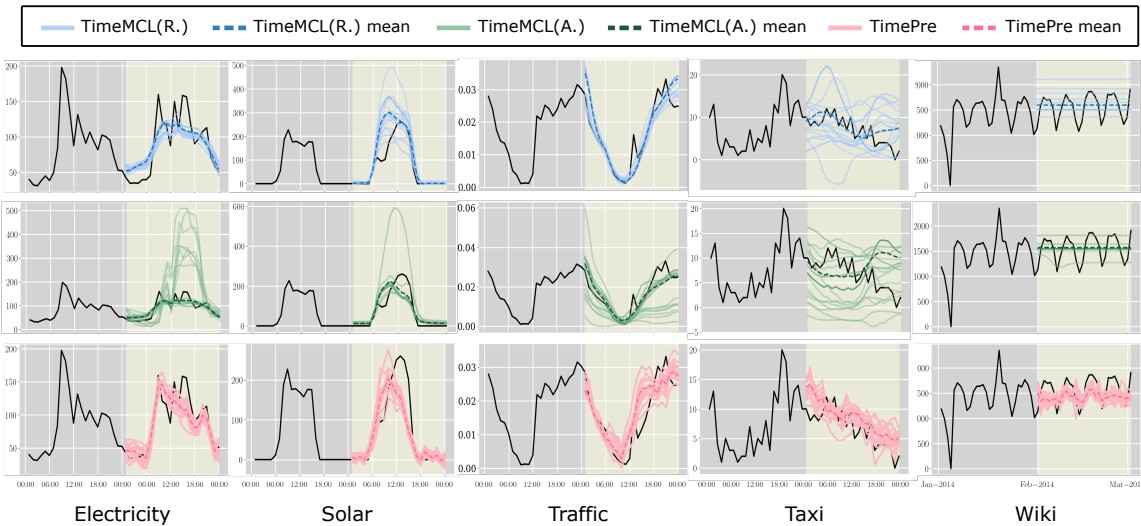

Figure 3: Qualitative forecasting results on five public data sets, comparing three models under the multi-hypothesis paradigm: TimeMCL (R.), TimeMCL (A.), and TimePre, all with 16 hypotheses.

**Visualization and Qualitative Analysis.** We qualitatively compare TimePre with TimeMCL on the Electricity, Solar, Traffic, Taxi, and Wiki data sets. As shown in Figure 3, TimePre produces more stable predictions, whereas the annealed variant exhibits scale drift and the relaxed variant shows only partial improvement. On Wiki, TimeMCL collapses into nearly constant outputs and fails to produce meaningful forecasts, while TimePre remains stable and captures realistic temporal variations.

## 4.3 Analysis of Different Normalizations

We analyze how the normalization layer affects stability and forecast quality under the MCL paradigm, and whether SIN's robust statistics improve hypothesis utilization over a non-robust reversible normalizer. The normalization comparison is averaged over five seeds; the calibration and head-utilization diagnostics are single-seed runs on the otherwise-identical TimePre architecture. All protocols are detailed in Appendix A.4.

**Normalization comparison.** We replace the Stabilized Instance Normalization module with four widely used normalization layers (BatchNorm (Ioffe & Szegedy, 2015), LayerNorm (Ba et al., 2016), GroupNorm (Wu & He, 2018), and InstanceNorm (Ulyanov et al., 2016)) and with the reversible instance normalization RevIN (Kim et al., 2022), holding all other components and the training protocol fixed. Table 3 and Figure 4 show that the choice of normalization is a decisive factor for training stability and representational quality. LayerNorm and GroupNorm are unstable during optimization, producing latent trajectories with inconsistent amplitude and severe

Table 3: Comparison of normalization methods on Electricity, averaged over five random seeds (mean $\pm$ standard deviation). CRPS-Sum values are multiplied by 100. Lower is better.

| Method | Distortion | CRPS-Sum |
|---|---|---|
| BatchNorm | $12447 \pm 442$ | $6.49 \pm 0.76$ |
| LayerNorm | $9453 \pm 334$ | $5.10 \pm 0.39$ |
| GroupNorm | $8452 \pm 523$ | $5.30 \pm 0.58$ |
| InstanceNorm | $9712 \pm 482$ | $6.23 \pm 0.84$ |
| RevIN | $8106 \pm 355$ | $3.26 \pm 0.23$ |
| SIN (Ours) | $\mathbf{7774 \pm 203}$ | $\mathbf{3.15 \pm 0.32}$ |

scale drift (Figure 4), while BatchNorm avoids divergence but, by aggregating statistics across a batch, disrupts the per-sample temporal structure and yields inferior accuracy. RevIN is the strongest baseline, yet SIN attains both lower Distortion (7774 vs. 8106) and lower CRPS-Sum (3.15 vs. 3.26), indicating that robust trimmed statistics provide a benefit beyond reversible normalization alone.

**Confidence Calibration Analysis.** Since the confidence head is trained with a winner-identification objective rather than a proper scoring rule, we evaluate its ranking quality using AUROC and argmax-winner accuracy, and compare uniform versus confidence-weighted CRPS surrogates (full protocol in Appendix A.4). Table 4 shows mixed behavior: confidence scores are useful on Traffic and Taxi, nearly equivalent to uniform weighting on Electricity and Wiki, and harmful on Exchange. We therefore do not interpret the confidence scores as fully calibrated probabilities; confidence calibration remains a limitation of the current model.

Table 4: Confidence-score diagnostics across six benchmarks. CRPS surrogate values are multiplied by 100; lower is better.

| Dataset | AUROC | Argmax acc. / Majority BL | Uniform CRPS | Conf. CRPS |
|---|---|---|---|---|
| Electricity | 0.489 | 0.000/0.286 | 3.17 | 3.17 |
| Exchange | 0.450 | 0.000/0.200 | 0.68 | 0.97 |
| Solar | 0.458 | 0.429/0.429 | 41.26 | 41.01 |
| Traffic | **0.833** | **0.857**/0.429 | 17.69 | 15.32 |
| Taxi | **0.786** | **0.589**/0.268 | 22.86 | 21.91 |
| Wiki | 0.600 | 0.000/0.200 | 6.22 | 6.23 |

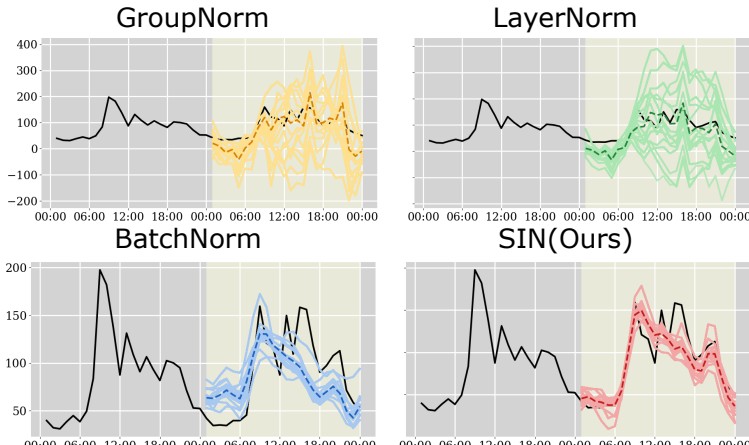

Figure 4: Forecasting comparison across normalization layers on Electricity. LayerNorm and GroupNorm are plotted on the range $[-200, 400]$, while BatchNorm and SIN are plotted on $[0, 200]$.

**Hypothesis Utilization.** To examine whether SIN mitigates hypothesis collapse, we analyze the winner-assignment distribution across the $K = 16$ heads (Tables 5 and 6; protocol in Appendix A.4). Read together with Distortion, head utilization exposes a failure mode the headline metric hides: BatchNorm, LayerNorm, GroupNorm, and InstanceNorm all reach moderate Distortion yet quietly collapse onto only 1–2 active heads. Only RevIN and SIN retain both low Distortion and head diversity, with SIN highest (2.29 bits, 6.0 active heads vs. RevIN's 1.66 bits, 5.0) and raising entropy over RevIN on all six benchmarks. This dissociation is precisely why a head-utilization diagnostic, beyond the headline Distortion, is necessary; Appendix E explains why SIN's robust statistics, rather than generic robustness, enlarge this utilization.

## 4.4 Isolation of the Failure Mode

To isolate the source of the instability diagnosed in Section 3.2, we vary the backbone and normalization on Electricity, with $K = 1$ (deterministic single-hypothesis regression, no WTA) and $K = 16$ (relaxed-WTA, $\varepsilon = 0.3$). As shown in Table 7, the Linear backbone with mean scaling already fails at $K = 1$, before any WTA competition; replacing mean scaling with RevIN or SIN restores normal performance under both backbones. This indicates that the failure is not intrinsic to lightweight backbones or to WTA alone, but arises from their interaction with inadequate normalization.

## 4.5 Ablation Study

**Effect of the Number of Hypotheses.** To evaluate the stability and scalability of TimePre, we vary the number of hypotheses $K$ on Electricity and compare against the Relaxed and Annealed TimeMCL variants.

As shown in Figure 5, TimePre remains the best-performing method across all tested values of $K$ and exhibits consistently smaller variance than the TimeMCL variants. Its Distortion generally improves as the number of hypotheses increases, suggesting that additional hypotheses are used effectively rather than causing unstable competition. In contrast, TimeMCL (R.) shows pronounced instability at $K=10$ and $K=12$, with large error bars and substantially worse Distortion. TimeMCL (A.) is more stable than TimeMCL (R.),

Table 5: Head-utilization diagnostic on Electricity under $K = 16$ relaxed-WTA training. Higher entropy and lower Gini indicate more balanced use of heads.

| Normalization | $H$ (bits) | Gini | Active heads |
|---|---|---|---|
| Mean | 0.000 | 1.000 | 1.0 |
| GroupNorm | 0.000 | 1.000 | 1.0 |
| InstanceNorm | 0.000 | 1.000 | 1.0 |
| BatchNorm | 0.246 | 0.995 | 2.0 |
| LayerNorm | 0.629 | 0.979 | 2.0 |
| RevIN | 1.660 | 0.873 | 5.0 |
| SIN (Ours) | **2.290** | **0.804** | **6.0** |

Table 6: SIN versus RevIN head-utilization entropy across six benchmarks. Entropy is measured in bits under $K = 16$ relaxed-WTA training.

| Dataset | RevIN $H$ | SIN $H$ | $\Delta H$ |
|---|---|---|---|
| Electricity | 1.66 | **2.29** | +0.63 |
| Traffic | 1.26 | **1.65** | +0.39 |
| Wiki | 3.32 | **3.55** | +0.23 |
| Solar | 1.32 | **1.51** | +0.19 |
| Exchange | 2.93 | **3.02** | +0.09 |
| Taxi | 3.57 | **3.58** | +0.01 |

Table 7: Controlled isolation study on Electricity. We vary the backbone, normalization strategy, and number of hypotheses. Lower is better.

| Backbone | Normalization | $K = 1$ (no WTA) | | $K = 16$ (relaxed-WTA) | |
|---|---|---|---|---|---|
| | | Distortion | CRPS-Sum | Distortion | CRPS-Sum |
| Linear | mean | 68670 | 47.20 | 52650 | 34.17 |
| Linear | RevIN | 8950 | 3.90 | 7800 | 3.13 |
| Linear | SIN | 9626 | 4.19 | **7524** | **2.98** |
| RNN (LSTM) | mean | 18000 | 7.05 | 10852 | 4.23 |
| RNN (LSTM) | RevIN | 11640 | 5.26 | 8237 | 3.38 |
| RNN (LSTM) | SIN | 10730 | 4.65 | 8072 | 3.46 |

but still remains consistently above TimePre (exact per-$K$ values are tabulated in Appendix D.1, Table 16). These results indicate that SIN helps stabilize multi-hypothesis learning as the number of competing heads increases.

**Effect of Different Backbones.** To test whether TimePre's gains depend on its backbone, we replace the single-layer linear backbone with three representative MLP-based models: DLinear (Zeng et al., 2023), TimeMixer (Wang et al., 2024), and TiDE (Das et al., 2023). Table 8 shows that a heavier backbone does not help and can hurt: the TiDE variant degrades sharply on Solar (478.26 vs. 267.06) with high variance. While TimeMixer gives the best Taxi result, the default linear backbone is best or second on Solar and Exchange and the most consistent overall, indicating that the gains stem mainly from the SIN preconditioning rather than backbone complexity.

Table 8: Distortion risk comparison on three data sets. Lower is better. Mean ± standard deviation over five runs.

| Model | Exchange | Solar | Taxi |
|---|---|---|---|
| TimePre | $0.0275 \pm 0.0004$ | $267.06 \pm 1.55$ | $182.27 \pm 1.86$ |
| TimePre (D.) | $\mathbf{0.0271 \pm 0.0005}$ | $267.21 \pm 2.79$ | $179.82 \pm 0.83$ |
| TimePre (M.) | $0.0311 \pm 0.0017$ | $\mathbf{262.07 \pm 5.55}$ | $\mathbf{169.22 \pm 2.00}$ |
| TimePre (T.) | $0.0412 \pm 0.0016$ | $478.26 \pm 73.69$ | $174.18 \pm 14.47$ |

**SIN with a Non-MLP Backbone.** To test whether the benefit of SIN is specific to the linear backbone, we equip the original LSTM backbone of TimeMCL with SIN, keeping $K = 16$ and the relaxed-WTA objective; comparing against TimeMCL (R.) isolates the effect of SIN under the same recurrent backbone. As shown in Table 9, SIN reduces Distortion on four of six benchmarks, with large gains on Electricity and Exchange and only small regressions on Solar and Traffic. The SIN-equipped LSTM is also competitive with TimePre, outperforming it on Taxi and Wiki. This indicates that the stabilizing effect of SIN is not restricted to the linear backbone used in TimePre.

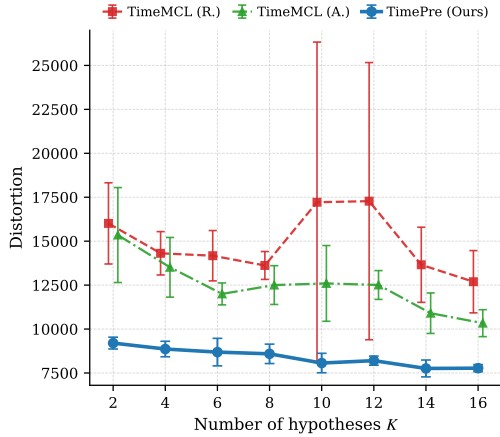

Figure 5: Effect of the number of hypotheses $K$ on Distortion on Electricity. Mean ± standard deviation over five runs.

Table 9: Effect of applying SIN to TimeMCL's original LSTM backbone. TimeMCL (R.) and TimePre are five-seed averages; SIN+TimeMCL is a single-seed diagnostic run. Lower is better.

| Dataset | TimeMCL (R.) | SIN + TimeMCL | TimePre (Ours) |
|---|---|---|---|
| Electricity | $12693 \pm 1772$ | 8072 | $7774 \pm 203$ |
| Exchange | $0.0380 \pm 0.0025$ | 0.0277 | $0.0275 \pm 0.0004$ |
| Solar | $292.15 \pm 11.68$ | 301.80 | $267.06 \pm 1.55$ |
| Traffic | $0.71 \pm 0.01$ | 0.728 | $0.68 \pm 0.02$ |
| Taxi | $191.23 \pm 5.34$ | 169.50 | $182.27 \pm 1.86$ |
| Wiki | $268832 \pm 9439$ | 249900 | $263492 \pm 2368$ |

Table 10: Temporal robustness under truncated forecast windows. We report per-step-normalized Distortion at 25%, 50%, 75%, and 100% of the prediction horizon. The ratio is $D_2(100\%)/D_2(25\%)$. Lower is better.

| Dataset | 25% | 50% | 75% | 100% | Ratio |
|---|---|---|---|---|---|
| Electricity | 3912.4 | 6198.6 | 6863.8 | 7622.8 | $1.95\times$ |
| Exchange | 0.0180 | 0.0222 | 0.0247 | 0.0277 | $1.54\times$ |
| Solar | 50.546 | 324.2 | 315.0 | 273.8 | $5.42\times$ |
| Traffic | 0.7185 | 0.5776 | 0.6527 | 0.7435 | $1.03\times$ |
| Taxi | 162.1 | 188.4 | 199.4 | 200.3 | $1.24\times$ |
| Wiki | 379,406 | 457,942 | 440,077 | 436,158 | $1.15\times$ |

**Temporal Robustness.** We further examine whether TimePre degrades sharply at later forecast steps. For each dataset, we evaluate a single model trained at the full prediction horizon on truncated forecast windows at 25%, 50%, 75%, and 100% of the horizon. This isolates lead-time degradation from changes in the context length, which is coupled to the prediction horizon in our training pipeline.

Table 10 shows that the far-horizon error remains stable on Traffic, Taxi, and Wiki, with degradation ratios between $1.03\times$ and $1.24\times$. Exchange and Electricity show moderate increases, while Solar exhibits a substantially larger and non-monotonic pattern ($5.42\times$). Solar is an hourly series dominated by a 24-hour day–night cycle, so the 25% window covers only a short low-variance nighttime segment and yields a small $D_2(25\%)$ baseline; extending the window to span the full diurnal cycle introduces the high-variance daytime portion, inflating both the error and the ratio. This also explains the non-monotonic profile, since the 25% window falls on the low-variance segment while wider windows capture the daily peak. Overall, this diagnostic suggests that TimePre does not show systematic runaway degradation on most datasets, although far-horizon robustness remains dataset-dependent.

**Hyperparameter Sensitivity.** We evaluate the sensitivity of TimePre to its three main hyperparameters on Electricity by varying one at a time while holding the others at their default values. Figure 6 summarizes the three sweeps for the relaxed-WTA coefficient $\varepsilon$, the confidence-loss weight $\beta$, and the SIN trimming ratio $p$ (exact values in Appendix D.1, Tables 13–15). In all three sweeps the default configuration sits at or close to the best setting on the tested grid, and performance degrades gracefully away from it, suggesting that the reported results do not rely on narrow per-dataset tuning.

## 5 Conclusion

We drew inspiration from recent MLP-based models in LTSF and extended their design to the MCL paradigm. To address the optimization instability that arises when lightweight linear architectures are trained under the MCL objective, we proposed the Stabilized Instance Normalization mechanism to harmonize feature scales and stabilize optimization. By integrating a linear backbone with a direct forecasting paradigm, TimePre achieves the best Distortion on all six benchmark datasets and the best or highly competitive CRPS-Sum on most benchmarks, while maintaining extremely fast inference, thereby bridging accuracy, efficiency, and stability.

**Limitations.** Our study has several limitations. First, the contribution is deliberately minimal: SIN is a robust variant of reversible instance normalization rather than a fundamentally new mechanism, and its advantage over RevIN is dataset-dependent rather than uniform. Relatedly, the instability we address is specific rather than universal: our isolation study (Section 4.4) shows that a linear backbone with mean

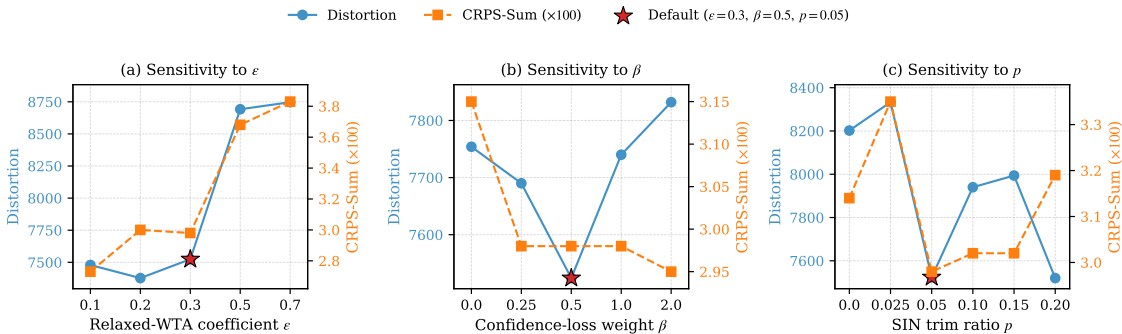

Figure 6: Hyperparameter sensitivity of TimePre on Electricity (single seed). Each panel sweeps one hyperparameter while holding the other two at their paper defaults. Blue (left axis, circles) is Distortion and orange (right axis, dashed squares) is CRPS-Sum ($\times 100$); lower is better for both. The red star marks the default ($\varepsilon$=0.3, $\beta$=0.5, $p$=0.05).

scaling already collapses at $K = 1$, before any WTA competition, whereas a recurrent backbone with mean scaling does not, so the failure stems from the particular combination of linear backbones, naive scaling, and WTA rather than from a general incompatibility between lightweight backbones and MCL. Second, the confidence head is trained with a binary winner-identification (cross-entropy) objective rather than a proper scoring rule; as our calibration analysis (Table 4) shows, the resulting scores are useful on some benchmarks but poorly calibrated on others, so we treat the confidence-weighted CRPS as a diagnostic rather than a guarantee of calibrated probabilities. Third, our evaluation focuses on six fixed-horizon GluonTS benchmarks, and the surrogate underlying SIN treats channels independently.

**Future work.** Several extensions follow naturally, all centered on probabilistic forecasting. First, the channel-independent design ignores cross-channel dependence; modeling the joint structure across variables while preserving the single-pass efficiency of MCL is a promising direction. Second, the unreliable confidence scores motivate calibration-aware training: wrapping the $K$ hypotheses in a conformal procedure would turn them into prediction regions with finite-sample coverage guarantees. A further step is to replace the winner-identification loss with a proper functional-quantization objective that jointly optimizes the hypotheses and their weights, which also raises the open question of how many hypotheses suffice to cover a $D$-dimensional future. Finally, validation on data with richer covariate structures and varying prediction horizons would strengthen the generality of our conclusions.

# 6 Broader Impact

Beyond the specific model, TimePre points to a broader direction for probabilistic forecasting. Sampling-based probabilistic forecasters represent uncertainty through repeated or iterative generation, which is accurate but costly; the multiple-choice-learning paradigm instead produces a full set of trajectory hypotheses in a single forward pass, reducing the energy and latency of uncertainty-aware forecasting by orders of magnitude. By showing that this one-pass paradigm can be made stable and competitive even with lightweight backbones, our work advances MCL as a practical, efficiency-oriented alternative for probabilistic forecasting, which we believe is increasingly important as such models are deployed at scale in resource-constrained or real-time settings. Concrete beneficiaries include renewable-energy and electricity-load management, traffic and mobility planning, and other operational decisions where forecast uncertainty matters but compute and latency are limited.

At the same time, probabilistic forecasts can be over-trusted. Since the confidence scores are not guaranteed to be well calibrated on every dataset, treating them as exact probabilities in high-stakes domains such as finance or critical infrastructure could lead to overconfident decisions. We therefore recommend validating calibration on the target domain and keeping a human in the loop for consequential decisions. Our experiments use no personal or sensitive data, and we foresee no risks beyond those common to time-series forecasting.

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

## A    Implementation Details

### A.1    Hyperparameters

Table 11 lists the key hyperparameters of TimePre used across all experiments. Unless otherwise stated, the same configuration is applied to all six benchmark datasets. For the main five-seed benchmark results, we follow the TimeMCL seed protocol and use the fixed seed set $\{3141, 3142, 3143, 42, 43\}$.

Table 11: Key hyperparameters of TimePre. All values are shared across the six benchmark datasets unless otherwise noted.

| Hyperparameter | Symbol | Value |
|---|---|---|
| Number of hypotheses | $K$ | 16 |
| Trimming ratio (SIN) | $p$ | 0.05 |
| Relaxed WTA coefficient | $\varepsilon$ | 0.3 |
| Confidence loss weight | $\beta$ | 0.5 |
| Trajectory head layers | – | 1 (linear) |
| Trajectory head hidden size | – | $H \times D$ |
| Confidence head layers | – | 2 (MLP) |
| Confidence head hidden size | – | 128 |
| Optimizer | – | Adam |
| Learning rate | – | $10^{-3}$ |
| Batch size | – | 200 |
| Batches per epoch | – | 30 |
| Training epochs | – | 200 |
| Early stopping patience | – | 10 epochs |
| Numerical stability constant | $\epsilon$ | $10^{-5}$ |
| Random seeds | – | $\{3141, 3142, 3143, 42, 43\}$ |

Unless otherwise stated, single-seed diagnostic experiments use seed 3141 and are marked as diagnostic runs in the corresponding captions.

### A.2    Runtime and FLOPs Measurement Protocol

All runtime measurements are conducted on a single NVIDIA RTX A6000 GPU with 48 GB memory. We report wall-clock inference time averaged over 100 forward passes after a 10-pass warm-up, using a fixed batch size of 200 samples. FLOPs are computed using `torch.profiler` with `profile_memory=False`, covering all linear, normalization, and activation operations in a single forward pass. Both metrics are measured on the Exchange dataset under identical conditions for all models.

### A.3    CRPS-Sum Computation

Our CRPS-Sum implementation follows the GluonTS evaluation protocol. For MCL-based models that produce $K$ discrete trajectory hypotheses rather than parametric distributions, we construct an empirical CDF from the $K$ predicted trajectories, weighted by the confidence scores $\{\gamma^{(k)}\}$. CRPS is then computed via the standard integral formulation applied to the aggregated (summed across variables) forecast at each horizon step.

### A.4    Diagnostic protocols

All diagnostics in Section 4.3 share TimePre's architecture and training protocol (single linear backbone, $K{=}16$, relaxed-WTA $\varepsilon{=}0.3$, 200 epochs, Adam at $10^{-3}$ with patience-10 early stopping), varying only the component under study. The normalization comparison (Table 3) is averaged over five random seeds; the confidence-calibration and head-utilization diagnostics are single-seed runs (seed 3141). Distortion and CRPS-Sum are reported, with CRPS-Sum scaled by 100.

**Normalization comparison (Table 3).**    We replace SIN with one of five alternatives (BatchNorm, LayerNorm, GroupNorm, InstanceNorm, RevIN), used as the per-instance input scaler, while holding all other components and the training protocol fixed, so that normalization is the sole varying factor.

**Confidence-calibration diagnostic (Table 4).** For each test instance we run the trained network directly—bypassing the sample-forecast generator, which normalizes the per-instance scores to sum to one— and read the raw post-sigmoid confidence $s_k \in [0,1]$ of each of the $K{=}16$ heads together with its predicted trajectory $\hat{F}_k$. The *winner* of an instance is the head closest to the ground truth, $k^\star = \arg\min_k \|\hat{F}_k - X\|^2$. We then compute three quantities. (i) **Per-head AUROC**: for each head, the tie-safe Mann–Whitney AUROC of $s_k$ against the winner indicator (average ranks via `scipy.stats.rankdata`), macro-averaged over heads with both classes present. (ii) **Argmax accuracy**: the fraction of instances on which $\arg\max_k s_k$ equals $k^\star$, compared against the chance rate $1/K$ and a majority-head baseline. (iii) **Weighted-quantile CRPS surrogate**: treating the $K$ trajectories as a weighted empirical predictive distribution, we compute the sum-CRPS over 19 quantile levels (with the gluonts factor of 2) under *uniform* weights $1/K$ versus *confidence* weights $s_k/\sum_j s_j$, and report the relative change. To avoid leakage we strip the last $T$ steps from every test series before feeding the model, mirroring `make_evaluation_predictions`; the recomputed Distortion matches the logged value within 1%.

**Head-utilization diagnostic (Tables 5 and 6).** At the end of training we record, on the validation set, the number of times each head $k$ is selected as the WTA winner, $n_k$, and form the winner distribution $\pi_k = n_k/\sum_j n_j$. We report three summaries: the Shannon entropy $H(\pi) = -\sum_k \pi_k \log_2 \pi_k$ in bits (maximized at $\log_2 16 = 4$ for uniform utilization), the Gini coefficient of $\{n_k\}$ (0 for perfectly balanced use, 1 for collapse onto a single head), and the number of active heads $|\{k : n_k > 0\}|$. Table 5 reports these per normalization on Electricity, and Table 6 reports RevIN-versus-SIN entropy on all six benchmarks.

# B  Dataset Details

We evaluate our method on six widely used probabilistic time-series forecasting benchmarks from the GluonTS library, namely *Solar*, *Electricity*, *Exchange*, *Traffic*, *Taxi*, and *Wikipedia*. All datasets contain strictly positive real-valued sequences and come with standard train–test splits defined in prior work. An overview of their main characteristics is provided in Table 12.

Table 12: Summary of the benchmark datasets used in our experiments. $N$ denotes the number of time series, $T$ the length of each series, and "Freq." the sampling frequency. The prediction horizon $H$ follows the standard settings in prior work.

| Dataset | $N$ | $T$ | Freq. | Horizon $H$ |
|---|---|---|---|---|
| Solar | 137 | 7009 | Hourly | 24 |
| Electricity | 370 | 5833 | Hourly | 24 |
| Exchange | 8 | 6071 | Daily | 30 |
| Traffic | 963 | 4001 | Hourly | 24 |
| Taxi | 1214 | 1488 | 30-min | 24 |
| Wikipedia | 2000 | 792 | Daily | 30 |

**Solar.** The Solar dataset contains hourly aggregated power production from 137 photovoltaic plants over roughly 7,000 time steps. The series exhibit strong daily seasonality induced by the day–night cycle, making them a canonical benchmark for modeling periodic but weather-dependent generation patterns.

**Electricity.** The Electricity dataset consists of hourly electricity consumption for 370 customers over 5,833 time steps. Demand typically follows both daily and weekly cycles driven by human activity and business operations, and it can be affected by holidays and load spikes, which pose challenges for probabilistic forecasting models.

**Exchange.** The Exchange dataset contains daily foreign exchange rates for 8 major currency pairs, each with 6,071 observations. Unlike energy or traffic data, these financial series seldom show clear periodicity; instead, they reflect macroeconomic conditions and market events, providing a non-seasonal and highly stochastic forecasting scenario.

**Traffic.** The Traffic dataset records road occupancy rates (bounded in $[0,1]$) from 963 loop sensors, sampled hourly for approximately 4,000 time steps. The series display pronounced rush-hour peaks as well as systematic differences between weekdays and weekends, which makes them a representative benchmark for high-dimensional, strongly seasonal traffic flows.

**Taxi.** The Taxi dataset is based on taxi ride counts in New York City, aggregated at 1,214 spatial locations every 30 minutes. We use the standard preprocessed version from GluonTS, which includes data from January 2015 (training) and January 2016 (testing). The resulting series capture complex spatial–temporal patterns and irregular spikes in demand.

**Wikipedia.** The Wikipedia dataset contains daily page-view counts for 2,000 popular Wikipedia pages. These series exhibit a mixture of long-term trends, weekly seasonality, and occasional bursts due to external events or campaigns. Following prior work, we adopt the official split and treat this dataset as a challenging benchmark for high-dimensional, event-driven demand forecasting.

Across all datasets, we follow the official train–test splits provided by GluonTS and prior benchmarks. For validation, we reserve the last few time steps before the forecast horizon within the training portion, as summarized in Table 12.

## C   Evaluation Metrics

We evaluate our approach using four metrics that capture both probabilistic forecasting quality and computational efficiency: Distortion, CRPS-Sum, FLOPs, and Runtime. Among them, Distortion serves as our primary evaluation metric.

**Distortion.** Distortion measures how well the set of predicted hypotheses covers the true target distribution. Given $K$ predicted trajectories $\{\hat{\mathbf{y}}^{(k)}\}_{k=1}^{K}$ for each ground-truth sequence $\mathbf{y}$, distortion is defined as the minimum Euclidean distance between the target and the closest hypothesis:

$$\text{Distortion}(\mathbf{y}) = \min_{1 \leq k \leq K} \left\| \mathbf{y} - \hat{\mathbf{y}}^{(k)} \right\|_2. \tag{9}$$

The final score is obtained by averaging over all test samples:

$$\text{Distortion} = \frac{1}{N} \sum_{i=1}^{N} \min_{1 \leq k \leq K} \left\| \mathbf{y}_i - \hat{\mathbf{y}}_i^{(k)} \right\|_2. \tag{10}$$

Lower distortion indicates better probabilistic coverage and sharper forecasting quality.

**CRPS-Sum.** The Continuous Ranked Probability Score (CRPS) (Gneiting & Raftery, 2007) evaluates the accuracy of a predictive distribution. Following prior work and the GluonTS implementation, CRPS-Sum is computed by first aggregating all individual time series and then applying CRPS to the resulting summed distribution at each forecast horizon. Formally, let $\tilde{y}_t = \sum_{d=1}^{D} y_{t,d}$ denote the aggregated target at time step $t$, and let $\tilde{F}_t$ be the predictive CDF of the corresponding aggregated forecast. CRPS-Sum is then defined as:

$$\text{CRPS-Sum} = \sum_{t=1}^{H} \text{CRPS}\big(\tilde{F}_t, \tilde{y}_t\big). \tag{11}$$

Lower CRPS-Sum indicates better calibrated probabilistic forecasts on the aggregated series.

**FLOPs.** Floating Point Operations measure the computational cost of a single forward pass. We compute FLOPs using the standard profiling tools in PyTorch, covering all linear, convolutional, normalization, and activation operations. Lower FLOPs indicate better computational efficiency and scalability.

**Runtime.** Runtime records the wall-clock time required for one forward pass on a single GPU under identical batch size and sequence length. This metric reflects real-world inference latency and complements FLOPs by capturing implementation overhead and hardware-level optimizations.

## D   Baseline Details

We compare our method against a wide range of strong probabilistic forecasting baselines, including both classical likelihood-based models and recent deep learning approaches. All baseline implementations follow the official code from GluonTS or the authors' releases, and we use the standard hyperparameters recommended in their original papers to ensure fair comparison.

Table 13: Sensitivity to the relaxed-WTA coefficient $\varepsilon$ on Electricity. Results are from a single seed, and CRPS-Sum values are multiplied by 100. The default value is in bold, and the best result is underlined. Lower is better.

| $\varepsilon$ | Distortion | CRPS-Sum |
|---|---|---|
| 0.10 | 7479 | 2.73 |
| 0.20 | 7377 | 3.00 |
| 0.30 | **7524** | **2.98** |
| 0.50 | 8692 | 3.68 |
| 0.70 | 8747 | 3.83 |

Table 14: Sensitivity to the confidence-loss weight $\beta$ on Electricity. Results are from a single seed, and CRPS-Sum values are multiplied by 100. The default value is in bold, and the best result is underlined. Lower is better.

| $\beta$ | Distortion | CRPS-Sum |
|---|---|---|
| 0.00 | 7754 | 3.15 |
| 0.25 | 7690 | 2.98 |
| 0.50 | **7524** | **2.98** |
| 1.00 | 7740 | 2.98 |
| 2.00 | 7832 | 2.95 |

**DeepAR.** DeepAR (Salinas et al., 2020) is an autoregressive probabilistic model based on LSTM networks. It predicts future values by estimating a parametric likelihood (e.g., Gaussian or Negative Binomial) and sampling from the learned distribution. As a widely adopted baseline, DeepAR captures temporal dependencies through recurrent structures but may struggle with long-term dependencies due to sequential recurrence.

**TimeGrad.** TimeGrad applies a conditional variational autoencoder (CVAE) framework to time-series forecasting. By learning latent stochastic dynamics through diffusion-based variational inference, it provides expressive probabilistic forecasts. Its training, however, requires sampling latent variables per time step, leading to slower inference.

**TimeMCL.** TimeMCL (Perera et al., 2024b) extends the Multiple Choice Learning (MCL) framework to time-series forecasting by training $K$ parallel prediction heads with a winner-takes-all assignment scheme. It generates diverse hypotheses that better cover the multimodal distribution of future trajectories. Although effective, the dense latent representations in TimeMCL lead to high cross-channel covariance and less stable training on some datasets.

**TimeMixer.** TimeMixer is a multi-period decomposition architecture that mixes temporal patterns across fine-to-coarse granularities. It performs especially well on seasonal datasets due to its frequency-aware decomposition and learned periodic mixing kernels.

**TiDE.** TiDE uses a two-stage architecture: an encoder that extracts temporal representations and a decoder that predicts the full future horizon in one shot. Its MLP-based structure allows for efficient training, although it can be sensitive to hyperparameter choices and normalization strategies.

**DLinear.** DLinear is a highly efficient linear modeling baseline that decomposes the input into trend and seasonal components using two simple linear layers. Despite its simplicity, it achieves strong performance on many long-term forecasting benchmarks and is widely used as a lightweight baseline.

For all baselines, we use the standard forecasting horizon defined in prior benchmarks and measure probabilistic performance using Distortion and CRPS-Sum (Section C). Computational efficiency is compared via FLOPs and Runtime under identical batch size and hardware settings.

## D.1 Detailed numerical results for the main-text figures

For completeness, this section tabulates the exact values underlying the hyperparameter-sensitivity and number-of-hypotheses figures in the main text (Figures 6 and 5). The hyperparameter sweeps in Tables 13, 14, and 15 vary one of $\varepsilon$, $\beta$, and the SIN trimming ratio $p$ at a time on Electricity while holding the others at their defaults; all numbers are from a single seed.

Table 15: Sensitivity to the SIN trim ratio $p$ on Electricity. Results are from a single seed, and CRPS-Sum values are multiplied by 100. The default value is in bold, and the best result is underlined. Lower is better.

| Trim ratio $p$ | Distortion | CRPS-Sum |
|---|---|---|
| 0.000 | 8202 | 3.14 |
| 0.025 | 8333 | 3.35 |
| 0.050 | **7524** | **2.98** |
| 0.100 | 7940 | 3.02 |
| 0.150 | 7994 | 3.02 |
| 0.200 | 7520 | 3.19 |

Table 16: Effect of the number of hypotheses on Distortion risk on the Electricity data set. Lower is better. Results are averaged over five runs.

| #Hypotheses | TimeMCL (R.) | TimeMCL (A.) | TimePre (Ours) |
|---|---|---|---|
| 2 | $16012 \pm 2310$ | $15349 \pm 2702$ | $\mathbf{9201 \pm 336}$ |
| 4 | $14311 \pm 1234$ | $13513 \pm 1698$ | $\mathbf{8864 \pm 441}$ |
| 6 | $14173 \pm 1432$ | $11999 \pm 624$ | $\mathbf{8688 \pm 783}$ |
| 8 | $13618 \pm 793$ | $12503 \pm 1104$ | $\mathbf{8590 \pm 552}$ |
| 10 | $17216 \pm 9112$ | $12597 \pm 2155$ | $\mathbf{8066 \pm 553}$ |
| 12 | $17277 \pm 7886$ | $12507 \pm 820$ | $\mathbf{8202 \pm 259}$ |
| 14 | $13657 \pm 2137$ | $10902 \pm 1152$ | $\mathbf{7758 \pm 480}$ |
| 16 | $12693 \pm 1772$ | $10335 \pm 767$ | $\mathbf{7774 \pm 203}$ |

Table 16 reports the per-$K$ Distortion on Electricity for TimePre and the Relaxed and Annealed TimeMCL variants, averaged over five runs; these are the values visualized in Figure 5.

# E  Why SIN Helps Winner-Takes-All Probabilistic Forecasting

Section 4.3 shows that SIN increases hypothesis utilization compared with RevIN. This cannot be explained only by saying that SIN is a more robust normalizer. The key question is how the input scale seen by the model affects the allocation of the $K$ hypotheses under the winner-takes-all (WTA) objective. This appendix studies this question through a simple product-source quantization model. The model suggests that scale heterogeneity across channels can concentrate the hypothesis budget on a few dominant channels, while leaving other channels with little or no probabilistic spread. We then show that the trimming step in SIN reduces this spurious heterogeneity relative to RevIN.

The two estimator-level results, Lemmas E.2 and E.3, are exact under their stated assumptions. Proposition E.1 uses a quantization surrogate and should be read as an explanatory model rather than as an exact characterization of the trained network.

Throughout this appendix, SIN and RevIN differ only in how they estimate the per-instance, per-channel location and scale. RevIN uses the empirical mean and variance of the context window. SIN uses their $p$-trimmed counterparts.

## E.1  WTA as $K$-point vector quantization

Let $X \in \mathbb{R}^D$ denote the normalized forecast target, where $D = T \cdot C$ is the flattened horizon-by-channel dimension and $P$ is the law of $X$. With hypotheses $\{F_k\}_{k=1}^K$, the relaxed-WTA training objective assigns each target mainly to the hypothesis with the smallest loss. Since the trajectory loss used in training is the squared $\ell_2$ winner loss, the corresponding population objective is the $K$-point quantization distortion

$$\mathcal{D}_K(P) = \min_{c_1,\ldots,c_K \in \mathbb{R}^D} \mathbb{E}_{X \sim P}\Big[\min_k \|c_k - X\|^2\Big]. \tag{12}$$

Under this view, the hypotheses act as codepoints of a vector quantizer for the predictive law, and a hypothesis is used when its codepoint receives a non-negligible part of the data distribution (Lee et al., 2016; Cortes et al., 2025).

The reported metric $D_2$ evaluates the same set of forecasts by the Euclidean distance $\|\hat{\mathbf{y}} - \mathbf{y}\|_2$ to the nearest hypothesis, rather than by the squared distance (Section C). Since $t \mapsto t^2$ is monotone on $t \geq 0$, the nearest-hypothesis assignment is the same under both criteria. The two objectives differ in the within-cell representative: the squared-error objective uses a conditional mean, whereas the Euclidean objective uses a geometric median. In the analysis below we use the squared-error objective $\mathcal{D}_K$, because it matches the loss optimized during training. We then interpret the resulting collapse behavior through $D_2$, which uses the same nearest-hypothesis assignment.

The relevant issue is therefore not only whether the codepoints reduce average error. It is also important to ask across which channels the codepoints are diversified, because this determines whether the $K$ forecasts cover the joint future or vary only along a small subset of channels.

## E.2  Scale heterogeneity can collapse the hypothesis budget

To make the allocation problem tractable, we approximate the post-normalization target by an independent product source $X = (X_1, \ldots, X_C)$. Channel $c$ is summarized by an effective scalar variance

$$v_c = \sigma_c^2 / \hat{\sigma}_c^2,$$

where $\sigma_c^2$ is the clean target variance and $\hat{\sigma}_c^2$ is the variance removed by the normalizer. This reduces each channel's $T$-dimensional block to one effective coordinate and ignores cross-channel dependence. This is only a modeling approximation. Instance normalization rescales channels, but it does not decorrelate them.

Under this surrogate, the optimal high-rate allocation of a rate budget $R = \log_2 K$ across the $C$ parallel Gaussian coordinates follows the classical reverse water-filling solution (Cover & Thomas, 2006; Gersho & Gray, 1992):

$$b_c = \tfrac{1}{2}\big[\log_2(v_c/\theta)\big]_+, \qquad \sum_c b_c = R, \tag{13}$$

where $\theta$ is the water level determined by the budget.

**Proposition E.1** (Channel collapse). *Under equation 13, any channel with $v_c \leq \theta$ receives $b_c = 0$. Its optimal codepoints all coincide at $\mathbb{E}[X_c]$, so the $K$ hypotheses are identical on that channel and provide no probabilistic spread there. Greater heterogeneity in $\{v_c\}$ can move more channels below the water level. In the extreme case where $v_c \leq v_1 \, 2^{-2R}$ for all $c > 1$, only channel 1 is active, since $\theta = v_1 \, 2^{-2R} \geq v_c$ for every $c > 1$.*

Proposition E.1 describes the failure mode captured by the surrogate model. When the normalized channels have very different effective variances, the optimal allocation can reduce average distortion by spending most of the hypothesis budget on the dominant channels. The remaining channels are then represented by a single shared value. In that case, the $K$ trajectories differ only in a low-dimensional subspace, even if the average distortion appears acceptable. We use this as a model of the behavior that a trained WTA network may approximate, not as a claim about its exact optimum.

### E.3 SIN increases the diversified subspace

We next show how SIN reduces the spurious channel heterogeneity that RevIN can introduce under spikes. Consider a spike-contamination model. For channel $c$, the context window contains clean observations drawn i.i.d. from a law $F_c$ with variance $\sigma_c^2$, together with possible upper and lower spikes. Each upper spike is strictly larger than every clean observation, and each lower spike is strictly smaller. This is a simplified model of sensor faults or impulse-like outliers.

**Lemma E.2** (Robustness of the trimmed scale). *Let $k = \lfloor pL \rfloor$ and suppose each tail contains at most $k$ spikes.*

  (i) *By the order separation assumption, the spikes are exactly the largest and smallest order statistics. They are removed by trimming. Thus $\hat{\sigma}_{\mathrm{SIN},c}^2$ is the sample variance of clean observations in the central block. It remains bounded for up to $k$ spikes per tail, with finite-sample explosion breakdown point $(k+1)/L$, which tends to $p$ as $L \to \infty$. If the spike fraction is $o(1)$, its large-sample limit is the variance functional of $F_c$ truncated to its central $1 - 2p$ mass. If the spike fraction does not vanish, the retained clean quantile interval can shift with the per-tail spike count.*

  (ii) *The empirical variance used by RevIN has explosion breakdown point $1/L \to 0$. A single spike of magnitude $M$ gives*
  $$\hat{\sigma}_{\mathrm{RevIN},c}^2 = \tfrac{M^2}{L}\left(1 - \tfrac{1}{L}\right) \to \infty$$
  *as $M \to \infty$.*

*Proof.* For (ii), place one value at $M$ and the remaining $L - 1$ values at 0. The empirical variance is

$$\hat{\sigma}_{\mathrm{RevIN},c}^2 = \tfrac{M^2}{L}(1 - \tfrac{1}{L}),$$

which diverges as $M \to \infty$. Thus one contaminated point is sufficient.

For (i), each upper spike is larger than all clean values and each lower spike is smaller than all clean values. Therefore the spikes occupy the largest and smallest order statistics. If each tail contains at most $k$ spikes, all spikes lie in the trimmed bands and are removed. The remaining central block $\{s_{(k+1)}, \ldots, s_{(L-k)}\}$ contains only clean draws. The resulting estimator is the sample variance of those clean observations. When the spike fraction is $o(1)$, this estimator converges to the variance of $F_c$ truncated to its central $1 - 2p$ mass. Boundedness is lost only when a tail contains $k + 1$ spikes, which gives the stated breakdown point.  $\square$

For comparison across channels, assume that the clean laws share a common shape, up to location and scale. Also assume that the upper- and lower-tail spike counts are either $o(L)$ or have the same limiting per-tail fractions across channels. Under these conditions, the retained clean quantile interval is the same across channels, and the trimmed-to-full variance ratio is a channel-independent constant. This setting isolates contamination as the source of cross-channel asymmetry. Since the effective dimension below is invariant to global rescaling, we work with post-normalization variances up to a common factor.

**Lemma E.3** (Homogenization). *Model channel-dependent contamination as inflating the empirical variance by a factor $1 + \tau_c$, where $\tau_c \geq 0$. Then, by Lemma E.2 and the common-shape assumption,*

$$v_c^{\mathrm{SIN}} \; \propto \; 1, \qquad v_c^{\mathrm{RevIN}} \; \propto \; (1 + \tau_c)^{-1}.$$

*Define the effective dimension as*

$$d_{\mathrm{eff}}(v) = \frac{\left( \sum_c v_c \right)^2}{\sum_c v_c^2} \in [1, C].$$

*Then*

$$d_{\mathrm{eff}}(v^{\mathrm{SIN}}) = C \; \geq \; d_{\mathrm{eff}}(v^{\mathrm{RevIN}}),$$

*with equality if and only if all $\tau_c$ are equal. If the contamination makes $\{v_c^{\mathrm{RevIN}}\}$ strictly more unequal in the majorization order, then $d_{\mathrm{eff}}$ strictly decreases.*

*Proof.* The quantity $d_{\mathrm{eff}}$ is invariant to a global rescaling of $v$, so the proportionality constants do not matter. By Cauchy–Schwarz,

$$\left( \sum_c v_c \right)^2 \leq C \sum_c v_c^2,$$

with equality if and only if $v$ is constant. SIN gives a constant vector and therefore attains $C$. If $w_c = v_c / \sum_j v_j$, then

$$d_{\mathrm{eff}} = \frac{1}{\sum_c w_c^2}.$$

This quantity is Schur-concave in $w$, so it does not increase under a more unequal perturbation. The RevIN vector $v_c \propto (1 + \tau_c)^{-1}$ is non-constant whenever the $\tau_c$ differ. $\square$

Combining the preceding results gives the intended mechanism. Under the common-shape assumption, SIN brings the normalized channel variances to a common level. In the surrogate allocation problem of equation 13, this keeps the channel variances from being separated by spike-driven estimation errors. RevIN, in contrast, can assign overly large empirical variances to contaminated channels. This reduces the effective variance of cleaner channels after normalization and can move them below the water level $\theta$. By Proposition E.1, those channels then receive no hypothesis budget in the surrogate model. SIN therefore increases the subspace over which the $K$ forecasts can differ.

### E.4 Empirical signature and scope

A larger diversified channel subspace $\{c : v_c > \theta\}$ gives the $K$ hypotheses more directions along which to separate. This should support more active heads. When the subspace is broad, the WTA winner assignments can spread over more hypotheses and the assignment entropy is higher. When the subspace collapses, the assignments concentrate on the few heads that tile the dominant channels.

The two quantities are not identical. The active channel set counts channels with positive rate in the surrogate model, whereas head entropy counts hypotheses with positive assignment mass in the trained network. We therefore treat the entropy measurements as supporting evidence, not as a direct estimate of $\theta$.

This account is consistent with Table 6. The SIN-over-RevIN entropy gain is largest on Electricity ($+0.63$ bits), a dataset with strong cross-channel scale disparity (Section 3.2). The gain is negligible on Taxi ($+0.01$ bits), where the channels are already more balanced and both normalizers give utilization close to the $\log_2 16$ ceiling.

**Scope.** Lemma E.2 is exact under the spike model. Lemma E.3 is exact under the stated common-shape and multiplicative-inflation assumptions. Proposition E.1 relies on the high-rate optimal-allocation regime and on the independent product-source approximation. The actual WTA model learns an unconstrained, non-product quantizer. Thus the proposition should be viewed as a mechanistic explanation rather than as an exact result for the trained network.

The rigorous part of the analysis is the first step of the mechanism: trimming prevents a small number of spikes from making the normalized channels strongly heterogeneous. Within the surrogate model, this

heterogeneity is exactly what causes a $K$-point quantizer to abandon lower-variance channels. The analysis is based on squared-error distortion, which is the loss optimized during WTA training. The reported Euclidean $D_2$ re-scores the same set of hypotheses. Since squaring is monotone, both criteria choose the same nearest hypothesis for each target. Therefore a collapsed channel, where the codepoints coincide and have zero spread, is detected in the same way by $D_2$. We do not claim that the $D_2$-optimal quantizer is the same as the squared-error quantizer. The reverse water-filling result is specific to the squared-error objective.

