# OpenReview forum: "TimePre: Bridging Accuracy, Efficiency, and Stability in Probabilistic Time-Series Forecasting"
_TMLR — Decision pending for TMLR_

### Review · Reviewer_SgBz · 2026-05-13

**Summary Of Contributions:**

The paper proposes TimePre, a probabilistic time-series forecasting framework that combines the speed of lightweight forecasting models with the flexibility of Multiple Choice Learning. TimePre leverages the Stabilized Instance Normalization (SIN), a channel-wise, reversible normalization layer intended to prevent hypothesis collapse when Multiple Choice Learning is paired with lightweight linear forecasting models.

The authors argue that directly combining linear backbones with winner-takes-all multi-hypothesis training leads to unstable optimization because scale differences across channels cause only a small subset of hypotheses to receive useful gradients. SIN addresses this by computing trimmed per-instance, per-channel statistics before forecasting, thereby reducing scale imbalance and improving robustness to outliers.

TimePre is evaluated on six GluonTS benchmark datasets: Electricity, Exchange, Solar, Traffic, Taxi, and Wikipedia. The paper reports that TimePre achieves the best Distortion score on all six datasets and competitive or best CRPS-Sum on most of them. It also reports inference-time advantages over sampling-based probabilistic models.

The writing requires a thorough review, as references are not using parentheses when needed, and the results in the main table do not use bold notation (even though the captions state that they do).

**Audience:**

Yes

**Audience Explanation:**

Time series forecasting is a topic of interest for many machine and deep learning practitioners.
At the same time, the paper contribution is very specific and might be difficult for the majority of the TMLR audience to understand its impact.

**Broader Impact Concerns:**

No concerns

**Claims And Evidence:**

Yes

**Claims Explanation:**

The empirical evaluation supports the claims about the effectiveness of the proposed approach. TimePre consistently achieves a good performance with limited computational resources with respect to the chosen baselines.

However, I am not an expert on the topic. I cannot properly judge the quality of the baselines used in the experiments.

**Requested Changes:**

- Tables should report best values in bold, as stated in the captions
- Citations should be properly formatted

---

> ### Author Response · Authors · 2026-06-07
> **Response to Reviewer SgBz**
>
> We thank the reviewer for the careful reading and for pointing out the presentation issues. We have revised the manuscript accordingly.
>
> **Point  1: Best values were not shown in bold in the main tables.**
>
> > "the results in the main table do not use bold notation (even though the captions state that they do)."
>
> We have fixed this formatting issue. In the revised manuscript, the best values are now shown in bold and the second best values are underlined consistently across the main Distortion table, the CRPS-Sum table, and the ablation tables.
>
> **Point  2: Citation formatting should be corrected.**
>
> > "references are not using parentheses when needed."
>
> We have revised the citation style throughout the manuscript. Parenthetical citations now use the appropriate author--year format, and textual citations are used only when the cited work is part of the sentence.
>
> **Point  3: The writing requires a thorough review.**
>
> We performed a broader writing and structure pass in addition to the formatting corrections above. In particular, we clarified the introduction and contribution statements, softened over-stated claims such as the original "fundamental incompatibility" framing, and reorganized the experimental analysis to make the evidence easier to follow. We also added new analysis sections on normalization, confidence calibration, hypothesis utilization, failure-mode isolation, robustness, and hyperparameter sensitivity.
>
> **Point  4: The contribution may be specific and its impact may be hard for the broader TMLR audience to see.**
>
> We have added a Broader Impact section and revised the introduction and conclusion to make the broader relevance clearer. The revised manuscript emphasizes that TimePre is not only a model-specific contribution, but also an attempt to make multi-hypothesis probabilistic forecasting more efficient and stable. We now discuss potential applications such as energy management, electricity-load forecasting, and traffic or mobility planning, as well as limitations related to calibration and deployment in high-stakes settings.
>
> **Point  5: Difficulty judging the quality of the baselines.**
>
> We clarified the baseline protocol in Section 4.1 and Appendix D. All baselines are reproduced in our own environment under a unified five-seed protocol, following the public TimeMCL reproduction scripts and the recommended settings of the corresponding original methods. The baseline set covers the main probabilistic forecasting paradigms, including MCL-based, copula-based, diffusion-based, flow-based, autoregressive, and classical forecasting methods.
>
> ---
>
> We thank the reviewer again for the careful reading and constructive suggestions. The requested formatting and citation fixes, together with the broader writing revisions, have made the manuscript clearer and easier to evaluate.

---

> > ### Comment · Reviewer_SgBz · 2026-06-08
> >
> > I thank the authors for addressing my concerns. I have no further comments.

---

### Review · Reviewer_i96o · 2026-05-20

**Summary Of Contributions:**

This paper presents TimePre, a framework that unites efficient linear/MLP forecasting backbones with Multiple Choice Learning (MCL) for probabilistic time-series forecasting. The contribution of TimePre is the development of a way to stabilize the disparity in scales from multiple channels during training for MCL by using trimmed statistics, thus preventing hypothesis collapse from occurring. Predictive performance comparisons indicate that TimePre significantly outperforms sampling-based models. Additionally, TimePre computes predictions faster than the previously mentioned sampling-based models.

Key strengths:
-  The paper precisely identifies the cause of hypothesis collapse (scale imbalance in linear backbones under MCL) and proposes a minimal, well-motivated fix: Stabilized Instance Normalization (SIN).
- The paper thoroughly evaluates TimePre through the use of multiple datasets with detailed reporting on normalized factors, performance, and the effect of using various normalization methods, forecasting backbone types, and number of hypotheses.

Key weaknesses:

- The core SIN mechanism is very similar to prior Reversible Instance Normalization (RevIN) [R1]. The paper fails to properly position itself against RevIN and other robust normalization alternatives.
- The paper also lacks calibration metrics to verify probabilistic quality and sensitivity analysis for key hyperparameters ( ε, β).


[R1] Reversible Instance Normalization for Accurate Time-Series Forecasting against Distribution Shift. ICLR 2022.

**Audience:**

Yes

**Audience Explanation:**

Although there are some improvements to be made in this paper (see Requested Changes), the primary contribution of this paper: the development of a new framework that addresses a stability issue in order to perform probabilistic forecasting more efficiently, is a good step for researchers studying time series forecasting, designing efficient models, and similar subjects. Thus, the findings of this paper should be of interest to at least some participants within TMLR's audience (for instance, those working on time series, generative models, efficient architectures, and normalization techniques).

**Broader Impact Concerns:**

The paper currently lacks a dedicated "Broader Impact" section. Thus adding such a section is strongly recommended​. This section should discuss both potential beneficial applications (e.g., in sustainable energy management) and the risks.

**Claims And Evidence:**

Yes

**Claims Explanation:**

The key findings stated in this submission, namely, that SIN removes a critical instability from MCLs using linear backbones, and that TimePre achieves very low distortion, a high CRPS-Sum, and good efficiency, are convincingly backed by substantial direct experimental results across many benchmark datasets. While the main limitations of the evidence are not based on evidence supporting the claim itself, they are based on the lack of completeness of the empirical studies and on putting the novel contributions into context with related work.

**Requested Changes:**

1. Include a full comparison to Reversible Instance Normalization (RevIN)​ [R1] to precisely capture all novel aspects of SIN, and clarify how SIN represents an incremental advancement over RevIN.

2. Carry out a comprehensive sensitivity analysis for the most important hyperparameter values (relaxation ε, loss weight β).

3. Provide additional analysis regarding temporal robustness by testing consistency in model performance across multiple prediction horizons and data types.

[R1] Reversible Instance Normalization for Accurate Time-Series Forecasting against Distribution Shift. ICLR 2022.

---

> ### Author Response · Authors · 2026-06-07
> **Response to Reviewer i96o(Part1)**
>
> We thank the reviewer for the positive assessment and for identifying two important weaknesses: the need to position SIN more clearly relative to RevIN, and the need for additional calibration and sensitivity analyses. We have revised the manuscript accordingly and address each change below.
>
> ---
>
> **Point 1: SIN is closely related to RevIN and should be positioned more precisely.**
>
> > "The core SIN mechanism is very similar to prior Reversible Instance Normalization (RevIN). The paper fails to properly position itself against RevIN and other robust normalization alternatives."
>
> We revised the related work, method, and limitation sections to position SIN as a robust extension of RevIN rather than as a fundamentally separate normalization mechanism. Specifically, we now narrow down that SIN follows RevIN's reversible normalization-and-denormalization structure, but replaces the empirical mean and variance with trimmed per-channel statistics to reduce the effect of outliers and scale spikes under WTA-style training.
>
> We also added direct RevIN comparisons in the experiments. In the revised normalization comparison, RevIN is included as a dedicated baseline under the same TimePre architecture and training protocol, and SIN achieves lower Distortion and CRPS-Sum than RevIN. We further added a head-utilization diagnostic reporting entropy, Gini coefficient, and active heads; SIN improves over RevIN on all three measures on Electricity. Finally, we extended the RevIN-vs-SIN entropy comparison to all six benchmarks and added Appendix E to explain, through a quantization-surrogate analysis, why trimmed statistics can improve hypothesis utilization under WTA training.
>
> ---
>
> **Point 2: The paper lacks calibration metrics for probabilistic quality.**
>
> > "The paper also lacks calibration metrics to verify probabilistic quality..."
>
> We clarified that our primary metric, Distortion, is an oracle/quantization error computed by taking the minimum distance over the $K$ predicted hypotheses, and therefore does not directly depend on confidence weights. Thus, the main hypothesis-coverage results are separate from the calibration quality of the confidence head. The confidence head is used only as an optional ranking or weighting component at inference time, not as the basis of our main claim.
>
> We also added a confidence-score diagnostic in Section 4.3, Table 4 with AUROC, argmax-winner accuracy against a majority-head baseline, and uniform versus confidence-weighted CRPS surrogates. The results are mixed: the scores show evidence of useful ranking on Traffic and Taxi, but are weak or harmful on other datasets. We therefore revised the manuscript to avoid interpreting the scores as fully calibrated probabilities, explicitly list calibration as a limitation, and identify proper-score-based confidence training as future work.
>
>
>
> ---
>
> **Point 3: Sensitivity analysis for the key hyperparameters is missing.**
>
> > "Carry out a comprehensive sensitivity analysis for the most important hyperparameter values (relaxation $\varepsilon$, loss weight $\beta$)."
>
> We added sensitivity analyses for the relaxed-WTA coefficient $\varepsilon$, the confidence-loss weight $\beta$, and the SIN trim ratio $p$. These are summarized in Figure 6, with exact numerical results reported in the appendix.
>
> The default configuration is either optimal or close to the best setting in the tested grids: $\varepsilon=0.3$ is within 2.0% of the best Distortion, $\beta=0.5$ gives the lowest Distortion, and $p=0.05$ is essentially tied with the best trim ratio. These results suggest that TimePre is not highly sensitive to narrow hyperparameter choices and that the reported gains do not come from per-dataset hyperparameter tuning.
>
> ---
>
> **Point 4: Additional analysis of temporal robustness is needed.**
>
> > "Provide additional analysis regarding temporal robustness by testing consistency in model performance across multiple prediction horizons and data types."
>
> We added a temporal-robustness diagnostic in Section 4.5, Table 10. For each dataset, we evaluate a single model trained at the full prediction horizon on truncated forecast windows at 25%, 50%, 75%, and 100% of the horizon, and report per-step-normalized Distortion. This design isolates lead-time degradation under a fixed trained model. Retraining separate models for each horizon would change the training setup itself, since the forecasting pipeline couples the context window, prediction length, sampling windows, and early stopping behavior.
>
> The results are mostly stable on Traffic, Taxi, and Wiki, with degradation ratios between $1.03\times$ and $1.24\times$. Exchange and Electricity show moderate increases, while Solar exhibits a larger and non-monotonic pattern, likely due to its strong day-night cycle. We therefore revised the manuscript to state that TimePre does not show systematic runaway degradation on most datasets.

---

> ### Author Response · Authors · 2026-06-07
> **Response to Reviewer i96o(Part2)**
>
> **Point 5: The paper lacks a Broader Impact section.**
>
> > "The paper currently lacks a dedicated 'Broader Impact' section... This section should discuss both potential beneficial applications (e.g., in sustainable energy management) and the risks."
>
> We added a dedicated Broader Impact section. It discusses potential beneficial applications of efficient probabilistic forecasting, including renewable-energy and electricity-load management, traffic and mobility planning, and other operational decisions where uncertainty estimates are useful but compute and latency are limited. It also discusses risks: probabilistic forecasts and confidence scores can be over-trusted, especially because the confidence scores in our current framework are not uniformly calibrated across datasets. We therefore caution against treating these scores as exact probabilities in high stakes domains without domain-specific validation, calibration checks, and human oversight.
>
> ---
>
> We thank the reviewer again for these constructive suggestions. The revised manuscript now positions SIN more carefully relative to RevIN, includes calibration and sensitivity diagnostics, and more clearly states both the potential impact and the limitations of the method.

---

### Review · Reviewer_VyJH · 2026-05-25

**Summary Of Contributions:**

The paper introduces TimePre, a probabilistic time-series forecasting framework using Multiple Choice Learning (MCL) with a lightweight linear/MLP-style backbone. Its core component is Stabilized Instance Normalization (SIN), which uses trimmed per-instance, per-channel statistics to reduce scale imbalance before winner-takes-all multi-hypothesis training. The paper reports best Distortion on six GluonTS benchmarks and best CRPS-Sum on five of six, while being faster than sampling-based baselines.

**Additional Comments:**

I am not an expert on the topic (I used to work on this topic but for several years I haven't been active). Therefore, my review confidence is limited (3/5).

**Audience:**

Yes

**Audience Explanation:**

Yes, the paper focus on time series forecasting, which is a common machine learning problem area. It is expected that numerous researchers working in this area would be interested in the paper findings.

**Broader Impact Concerns:**

Not applicable.

**Claims And Evidence:**

No

**Claims Explanation:**

The motivation behind the proposed method is clear: Sampling-based probabilistic forecasters such as TimeGrad, TempFlow, and TACTiS-2 can be computationally expensive, while MCL-style forecasting gives a finite set of future trajectories in one pass. The paper’s attempt to make this efficient using a lightweight backbone is reasonable. The empirical results are strong on the paper’s primary metric. In Table 1, TimePre obtains the best Distortion on all six datasets, with large improvements over TimeMCL variants on Electricity, Exchange, Solar, Taxi, and modest gains on Wiki. The reported variance is also much lower than TimeMCL in several cases, supporting the stability claim.
The runtime/FLOPs performance is also very good. According to Figure 2, TimePre appears to be faster than all compared models, with very low FLOPs while also achieving the lowest Distortion. The normalization ablation is useful. Table 3 shows SIN outperforming BatchNorm, LayerNorm, GroupNorm, and InstanceNorm on Electricity, with both lower Distortion and CRPS-Sum. This gives some evidence that the normalization choice matters, not just the architecture.

My biggest concern is that the paper’s contribution and novelty is somewhat overstated. SIN can be thought of as a robust per-instance, per-channel normalization using trimmed mean/variance plus inverse scaling. While this is certainly a smart engineering choice, the paper presents it as resolving a “fundamental incompatibility” between lightweight backbones and MCL. The evidence for such a fundamental incompatibility is not strong enough. The paper mostly shows that TimeMCL variants behave worse than TimePre, but it does not sufficiently isolate whether the failure comes from linear backbones, WTA training, normalization, architecture capacity, implementation details, or hyperparameters.

The evaluation relies heavily on Distortion, which is aligned with the MCL training objective. This can favor models that output a diverse set of trajectories without necessarily producing well-calibrated probabilistic forecasts. CRPS-Sum partly addresses this, but Table 2 includes only four methods, omitting several baselines from Table 1. This makes the probabilistic-quality comparison incomplete.

I also have some subtle but potentially important methodological concerns: a) For MCL models, the paper constructs an empirical CDF from K hypotheses weighted by confidence scores, but the confidence calibration objective is only a binary winner-identification loss, not a proper likelihood or proper scoring rule. It is unclear whether the confidence scores are calibrated enough to justify weighted empirical CRPS. More calibration diagnostics would be needed. b) The baseline comparison should be further elaborated. The paper states that all baselines are reproduced under identical configurations and training settings, but it is unclear whether the competing methods are tuned comparably to TimePre.

The ablation on backbones is inconclusive. Table 5 evaluates TimePre variants on three datasets, and the results are mixed: TimeMixer is best on Taxi, DLinear-style default is best or near-best elsewhere, TiDE is unstable on Solar. This puts into question the claim that SIN bridges lightweight backbones and MCL broadly.

**Requested Changes:**

In addition to addressing the concerns in my comments above, I would be interested in the following questions:
- How many heads are active during training and inference? Does SIN increase active-head entropy compared with InstanceNorm and standard z-score normalization?
- How sensitive are results to the trimming ratio p?
- Why is CRPS-Sum reported for only four methods in Table 2, while Distortion is reported for nine?
- Are the confidence scores calibrated? It would be helpful to report reliability/calibration diagnostics or compare uniform-weight vs confidence-weighted empirical CRPS.
- Does SIN still help when applied to TimeMCL’s original backbone, or is it specific to the proposed architecture?

On a more minor note, I think that the citation style used is wrong (seems like the authors use \citet instead of \cite).

In terms of presentation, I think the Figure 1 is not so helpful as an introductory figure. I would much rather prefer a conceptual depiction of how TimePre works.

---

> ### Author Response · Authors · 2026-06-07
> **Response to Reviewer VyJH(Part1)**
>
> We thank the reviewer for the careful and constructive review. We agree that the original submission framed the observed failure mode too broadly and did not sufficiently isolate the factors contributing to it. This feedback directly led us to the revised claim framing and the additional controlled analyses in the manuscript. We respond point by point below.
>
> ---
>
> **Point 1: The contribution and novelty were over-stated, and the failure mode was not sufficiently isolated.**
>
> > "the paper presents it as resolving a 'fundamental incompatibility' ... it does not sufficiently isolate whether the failure comes from linear backbones, WTA training, normalization, architecture capacity, implementation details, or hyperparameters."
>
> We removed the original "fundamental incompatibility" framing and added a controlled isolation study in Section 4.4, Table 7. Holding the training protocol fixed, this study varies three factors on Electricity: the backbone (Linear vs. LSTM), the normalization strategy (mean scaling, RevIN, SIN), and the number of hypotheses ($K=1$ deterministic regression vs. $K=16$ relaxed-WTA).
>
> The results support a more specific interpretation of the failure mode. Linear+mean scaling is already highly unstable at $K=1$, before WTA competition is introduced, while replacing mean scaling with RevIN or SIN substantially restores performance. RevIN and SIN keep the K = 16 setting substantially more stable. This suggests that the issue is not caused by lightweight backbones or WTA alone, but by their interaction with inadequate normalization and scale handling.
>
> We therefore revised the abstract, introduction, contribution statements, and limitations to describe the issue more precisely, rather than presenting it as a universal incompatibility between lightweight forecasting models and MCL.
>
> ---
>
> **Point 2: The evaluation relied too heavily on Distortion, and the CRPS-Sum comparison was incomplete.**
>
> > "The evaluation relies heavily on Distortion ... Table 2 includes only four methods, omitting several baselines from Table 1. This makes the probabilistic-quality comparison incomplete."
>
> In the revision, Table 2 reports CRPS-Sum for all nine methods, matching the scope of the Distortion table. The expanded comparison gives a more balanced picture: TimePre achieves the best CRPS-Sum on Electricity, Exchange, and Wiki, and the second-best result on Solar and Taxi, while TimeGrad obtains the best CRPS-Sum on Solar, Traffic, and Taxi.
>
> We therefore narrow down the CRPS-Sum claims and distinguish the two metrics more clearly. Distortion evaluates hypothesis coverage and does not use the confidence scores, while CRPS-Sum evaluates distributional quality and is more sensitive to calibration. The revised results also clarify the trade-off between probabilistic quality and inference cost: TimeGrad can achieve stronger CRPS-Sum on some datasets, but it is a sampling-based diffusion model with substantially higher inference cost, whereas TimePre produces all hypotheses in a single forward pass.
>
> **Point 3: Confidence scores were used for weighted empirical CRPS without calibration evidence.**
>
> > "the confidence calibration objective is only a binary winner-identification loss, not a proper likelihood or proper scoring rule ... More calibration diagnostics would be needed."
>
> We clarify that our primary metric, Distortion, is computed by taking the minimum distance over the $K$ predicted hypotheses and does not use confidence weights. Thus, the main hypothesis coverage result is separate from the quality of the confidence head.
>
> We also added confidence score diagnostics in Section 4.3, Table 4, including AUROC, argmax winner accuracy, and CRPS diagnostics computed with uniform weights versus confidence weights. The results show a mixed picture: the confidence scores provide useful ranking signals on Traffic and Taxi, but are less reliable or even detrimental on other datasets. We therefore no longer interpret these scores as calibrated probabilities. In the revised manuscript, confidence based CRPS results are treated only as diagnostics, and calibration with proper scoring rules is listed as an important direction for future work.

---

> ### Author Response · Authors · 2026-06-07
> **Response to Reviewer VyJH(Part2)**
>
> **Point 4: How many heads are active, and does SIN increase active-head entropy?**
>
> > "How many heads are active during training and inference? Does SIN increase active-head entropy compared with InstanceNorm and standard z-score normalization?"
>
> We added hypothesis utilization diagnostics in Section 4.3, Tables 5 and 6. Specifically, we measure the validation set WTA winner-assignment distribution after training and report entropy, Gini coefficient, and the number of active heads. On Electricity under $K=16$, mean scaling, GroupNorm, and InstanceNorm collapse to one active head, while RevIN and SIN maintain more diverse winner assignments. SIN gives the highest entropy, the lowest Gini, and the largest number of active heads among the tested normalizers.
>
> We also extended the compare between RevIN and SIN to all six benchmarks. SIN increases winner-assignment entropy over RevIN on every dataset, with the largest gain on Electricity and the smallest gain on Taxi. In addition, we added Appendix E to explain this behavior through a quantization-surrogate analysis: scale heterogeneity can concentrate the hypothesis budget on a few dominant channels, while SIN's trimmed statistics reduce the spurious variance heterogeneity that can remain under RevIN. Since TimePre emits all $K$ hypotheses at inference time, these diagnostics focus on whether the trained heads are actually used by the WTA assignment. The results are consistent with the interpretation that SIN helps relaxed-WTA training maintain more balanced hypothesis utilization.
>
> ---
>
> **Point 5: How sensitive are results to the trimming ratio $p$?**
>
> > "How sensitive are results to the trimming ratio $p$?"
>
>
> We added a sweep over the trimming ratio on Electricity, summarized in Figure 6 with exact values reported in Appendix D.1. We evaluate $p \in {0, 0.025, 0.05, 0.10, 0.15, 0.20}$ while keeping the other hyperparameters fixed.
>
> The default $p=0.05$ gives near-best Distortion and the best CRPS-Sum in the tested grid. The $p=0$ case, which removes robust trimming and uses untrimmed instance statistics, performs worse than the default, suggesting that trimming contributes to the stability of SIN. Overall, performance does not collapse across the positive trim ratios tested, so the results do not appear to depend on a narrowly tuned value of $p$.
>
>
> ---
>
> **Point 6: The backbone ablation was inconclusive, questioning whether SIN broadly bridges lightweight backbones and MCL.**
>
> > "The ablation on backbones is inconclusive ... This puts into question the claim that SIN bridges lightweight backbones and MCL broadly."
>
> We revised the interpretation of the backbone ablation in Table 8. The comparison is intended to test whether the gains come simply from using a stronger backbone. The results do not support this explanation: replacing the default linear backbone with DLinear, TimeMixer, or TiDE does not consistently improve performance, and TiDE degrades sharply on Solar with high variance. We therefore now draw a narrower conclusion: the improvement is better attributed to SIN preconditioning than to increased backbone complexity. We also removed broad claims that SIN universally bridges arbitrary lightweight backbones and MCL, and only keep the supported claim that SIN also helps when applied to TimeMCL's recurrent backbone, as shown in Table 9.
>
>
> ---
>
> **Point 7: Does SIN still help when applied to TimeMCL's original backbone?**
>
> > "Does SIN still help when applied to TimeMCL's original backbone, or is it specific to the proposed architecture?"
>
> We added this diagnostic experiment in Section 4.5, Table 9. We equip TimeMCL's original LSTM backbone with SIN while keeping $K=16$ and relaxed-WTA training. This directly tests whether the effect of SIN is tied only to the proposed linear TimePre architecture.
>
> In this diagnostic run, SIN+TimeMCL reduces Distortion on four of six benchmarks compared with TimeMCL (R.), with clear improvements on Electricity and Exchange and additional gains on Taxi and Wiki. It shows small regressions on Solar and Traffic. Compared with TimePre, the SIN-equipped LSTM backbone is also competitive and performs better on Taxi and Wiki. These results suggest that the stabilizing effect of SIN is not limited to the linear backbone.

---

> ### Author Response · Authors · 2026-06-07
> **Response to Reviewer VyJH(Part3)**
>
> **Point 8: Are the competing methods tuned comparably to TimePre?**
>
> > "it is unclear whether the competing methods are tuned comparably to TimePre."
>
> We clarified the training and tuning protocol in Section 4.1, Appendix A and D. For the baselines, we follow the public TimeMCL reproduction scripts and the recommended settings of the corresponding original methods, and report the reproduced main results under the same five-seed protocol. TimePre uses one default configuration across all six benchmarks, with no separate tuning for each dataset.
>
> We also added sensitivity analyses for the relaxed-WTA coefficient $\varepsilon$, the confidence-loss weight $\beta$, and the trim ratio $p$, with exact values reported in Appendix D.1. The default configuration is at or near the best tested grid settings, which suggests that the reported gains are not driven by dataset specific hyperparameter search.
>
> ---
>
> **Point 9: Citation style.**
>
> > "the citation style used is wrong (seems like the authors use \citet instead of \cite)."
>
> We fixed the citation style throughout the manuscript. Parenthetical references now use the appropriate author year format, and textual citations are used only when the cited work is part of the sentence.
>
> ---
>
> **Point 10: Figure 1 is not helpful as an introductory figure.**
>
> > "Figure 1 is not so helpful as an introductory figure. I would much rather prefer a conceptual depiction of how TimePre works."
>
> We agree that the original introduction did not explain the mechanism of TimePre clearly enough before presenting the empirical comparison. In the revision, we strengthened the conceptual exposition in the introduction and method sections. In particular, Section 3.3 now presents TimePre as a three-stage pipeline consisting of SIN, a lightweight temporal encoder, and a multi-hypothesis decoder, and explains how SIN preconditions the input before WTA-style multi-hypothesis training.
>
> We retained the opening performance figure only as an empirical overview because it compactly summarizes the benchmark-level motivation across the six datasets. We no longer rely on it as the primary explanation of the method. Instead, the revised surrounding text introduces the pipeline and the role of SIN before the experimental results are discussed.
>
> ---
>
> We thank the reviewer again for the detailed and constructive feedback. The revised manuscript now provides a narrower claim, a controlled failure mode analysis, expanded CRPS-Sum evaluation, confidence score diagnostics, and hypothesis utilization results, which together make the scope and evidence of the paper clearer.

---

### Author Response · Authors · 2026-06-07
**Summary of the Authors' Responses**

We appreciate the reviewers' recognition of the motivation behind TimePre, especially the goal of combining the efficiency of lightweight forecasting backbones with the distributional flexibility of Multiple Choice Learning (MCL) for probabilistic time-series forecasting. We also thank the reviewers for helping us identify limitations in the original submission, including overly broad claim framing, insufficient isolation of the failure mode, incomplete CRPS-Sum reporting, limited calibration evidence, unclear positioning relative to RevIN, and missing robustness and broader-impact analyses.

To address these comments, we revised both the experiments and the manuscript as follows:

* **Claim Scope and Failure Mode (R-VyJH):** We removed the original "fundamental incompatibility" framing and now describe the problem more narrowly as an interaction among naive normalization, lightweight forecasting backbones, and WTA training with multiple hypotheses. We also added a controlled isolation study to separate the effects of backbone choice, normalization, and the number of hypotheses.

* **Probabilistic Evaluation and Calibration (R-VyJH, R-i96o):** We expanded the CRPS-Sum comparison to cover all methods in the main benchmark table, so that the probabilistic evaluation is no longer limited to a small subset of baselines. We also added diagnostics for the confidence scores and revised the manuscript to avoid treating these scores as calibrated probabilities.

* **SIN and Its Relation to RevIN (R-i96o, R-VyJH):** We revised the related work, method, experiments, and limitations to position SIN as a robust extension of RevIN rather than as a separate normalization principle. We added RevIN as a direct baseline and clarified that SIN keeps the reversible structure of RevIN while using trimmed statistics for each channel.

* **Hypothesis Utilization and Mechanistic Analysis (R-VyJH, R-i96o):** We added diagnostics for how the prediction heads are used under WTA training, including entropy, Gini coefficient, and the number of active heads. We also added a mechanistic explanation in Appendix E to explain why robust normalization can help maintain more balanced use of the hypotheses.

* **Robustness and Sensitivity Analyses (R-i96o, R-VyJH):** We added sensitivity analyses for the main hyperparameters, corrected the implementation-details table to match the reported experimental setting, tested SIN on the original LSTM backbone of TimeMCL, and added a truncated-horizon robustness analysis using 25%, 50%, 75%, and 100% forecast windows.

Furthermore, we addressed additional presentation and scope issues raised by the reviewers:

* **Training and Tuning Protocol (R-VyJH):** We clarified that the baselines follow the public TimeMCL reproduction scripts and the recommended settings of the corresponding methods, while TimePre uses one default configuration across all six benchmarks without tuning separately for each dataset.

* **Presentation, Citations, and Formatting (R-SgBz, R-VyJH):** We fixed citation formatting, corrected bold and underline notation in the main tables, and improved the explanation of TimePre as a three-stage pipeline consisting of SIN, a lightweight temporal encoder, and a decoder that produces multiple hypotheses.

* **Broader Impact (R-i96o):** We added a dedicated Broader Impact section discussing potential applications in energy, traffic, and other settings where efficient probabilistic forecasting is useful, as well as risks from over-trusting probabilistic forecasts and uncalibrated confidence scores.

Overall, we have added targeted diagnostics, expanded the probabilistic-quality evaluation, and clarified the role of SIN relative to RevIN. We sincerely thank all reviewers and the area chair for their time and feedback.